# Multiplatform analyses reveal distinct drivers of systemic pathogenesis in adult versus pediatric severe acute COVID-19

The pathogenesis of multi-organ dysfunction associated with severe acute SARS-CoV-2 infection remains poorly understood. Endothelial damage and microvascular thrombosis have been identified as drivers of COVID-19 severity, yet the mechanisms underlying these processes remain elusive. Here we show alterations in fluid shear stress-responsive pathways in critically ill COVID-19 adults as compared to non-COVID critically ill adults using a multiomics approach. Mechanistic in-vitro studies, using microvasculature-on-chip devices, reveal that plasma from critically ill COVID-19 adults induces fibrinogen-dependent red blood cell aggregation that mechanically damages the microvascular glycocalyx. This mechanism appears unique to COVID-19, as plasma from non-COVID sepsis patients demonstrates greater red blood cell membrane stiffness but induces less significant alterations in overall blood rheology. Multiomics analyses in pediatric patients with acute COVID-19 or the post-infectious multi-inflammatory syndrome in children (MIS-C) demonstrate little overlap in plasma cytokine and metabolite changes compared to adult COVID-19 patients. Instead, pediatric acute COVID-19 and MIS-C patients show alterations strongly associated with cytokine upregulation. These findings link high fibrinogen and red blood cell aggregation with endotheliopathy in adult COVID-19 patients and highlight differences in the key mediators of pathogenesis between adult and pediatric populations.

The spectrum of clinical disease resulting from SARS-CoV-2 infection is broad, ranging from mild to severe COVID-19 in symptomatic adults and including a post-infectious multisystem inflammatory syndrome in children (MIS-C). Large-scale epidemiologic studies demonstrate that up to 5% of unvaccinated, symptomatic adults with COVID-19 develop critical illness, with affected patients experiencing acute respiratory distress syndrome (ARDS), multisystem organ failure, and sometimes death[1]. The rate of severe COVID-19 is much lower in children; nevertheless, 10-33% of those requiring hospitalization are critically ill[2–4]. While the primary clinical manifestations of COVID-19 reflect infected epithelium of the upper and lower respiratory tracts, multi-organ involvement is common in severely ill adult patients[5,6]. In addition, children with post-infectious MIS-C experience systemic manifestations, which may occur after mild or even asymptomatic infection[7–9]. Previous studies generating proteomic, metabolomic, and transcriptomic data from patients with COVID-19 have uncovered alterations in numerous biological pathways associated with clinical severity[10–14]. Nevertheless, to our knowledge, pairing of multiomics datasets with mechanistic studies to uncover mediators of severe disease in both adult and pediatric patients has not been performed.

Increasing evidence suggests much of the organ dysfunction associated with severe COVID-19 results from widespread endothelial dysfunction and microvascular thrombosis[15–17]. The etiology of microthrombi appears multifactorial, with descriptions of hypercoagulability resulting from elevations in prothrombotic factors, like the hepatic acute phase protein fibrinogen and endothelial-derived factor

✉ e-mail: wilbur.lam@emory.edu; eortlun@emory.edu; cheryl.maier@emory.edu

VIII and von Willebrand factor, as well as blood hyperviscosity[18–20]. Furthermore, endothelial damage and vasculopathy appear to be exacerbated by the formation of neutrophil extracellular traps (NETs), complement activation, and increases in inflammatory cytokines[15,16,21–24]. The extent of these features may be unique to SARS-CoV-2 infection, as autopsy series have demonstrated severe endothelial injury and significantly more alveolar capillary microthrombi in patients expiring from COVID-19 compared to those expiring from H1N1 influenza[25]. Although endothelial cells express ACE2, the receptor used by SARS-CoV-2 for cellular entry, evidence supporting the presence of virus particles in the microvasculature that would account for the degree of systemic endotheliopathy has been limited and remains controversial. Recent data instead suggest endothelial damage arises from indirect processes related to complement and immune-mediated pathways resulting in microvascular injury[26].

Blood rheology, the behavior of blood flow at the cellular level, is a major determinant of endothelial health and function, and biophysical changes in blood flow can cause pathologic endothelial damage and associated microvascular clotting. Red blood cell (RBC) aggregation and membrane stiffness are major determinants of blood flow, and others have correlated these parameters with mortality in critically ill patients prior to the COVID-19 pandemic[27,28]. In addition, intravital studies using sublingual darkfield microscopy in critically ill patients have shown altered microvascular perfusion patterns and an associated volume loss of the endothelial glycocalyx, the proteoglycan complex involved in maintaining endothelial health[29]. However, the phenomenon remains understudied and has not been reported in patients with SARS-CoV-2 infection. Targeting the pro-inflammatory and pro-thrombotic factors in the plasma of critically ill COVID-19 patients has been identified as a potential therapeutic intervention[30], yet the influence of altered plasma components on the microvascular and endothelial biomechanical environment in COVID-19 remain unexplored.

Here, we undertook a combined multiomics and microfluidics-based approach to better understand the plasma and biophysical factors contributing to endothelial dysfunction and systemic disease in COVID-19. Through targeted and untargeted proteomics, lipidomics and metabolomics analysis, we identify complement and coagulation pathways as common elements in both adult and pediatric populations, and a unique pathway in adults related to fluid shear stress. Microfluidics analysis modeling the hemodynamic environment of the microvasculature and incorporating human endothelial cells and blood components suggests this pathway reflects changes in the biophysical properties of blood related to fibrinogen-induced RBC aggregation that directly damages the endothelial glycocalyx. Notably, this mechanism appears distinct from the microvascular damage related to increased RBC membrane rigidity observed in patients with non-COVID sepsis. We propose that alterations in the biomechanical properties of blood are an important contributor to the widespread endotheliopathy and organ dysfunction observed in adults with severe COVID-19. In contrast, integrative cytokine and multiomics analysis demonstrate little overlap between adult and pediatric cohorts, underscoring the role of immune dysregulation in pediatric disease and the variable nature of disease manifestations between these populations.

## Results

### Integrated multiomics analysis identifies alterations in coagulation and fluid shear stress response pathways in adults with COVID-19

Given the systemic disease manifestations observed in COVID-19, we hypothesized that characterization of the plasma lipidome, metabolome and proteome from severely ill patients might inform our understanding of mechanisms driving disease pathogenesis. We were most interested in identifying disease-related pathways unique to COVID-19 as compared to non-COVID severe illness. To do so, plasma was obtained from critically ill PCR-confirmed COVID-19-positive patients (COVID +, $n = 15$) and from critically ill PCR-confirmed COVID-19-negative patients (COVID-, $n = 10$) for multiomics investigation (Fig. 1A, Fig. S1A). Notably, all patient samples in this adult cohort were obtained on a single day in April 2020, before the use of now standard therapies or the availability of vaccines. Patient demographics and relevant clinical characteristics are provided in the Supplementary Information (Table S1).

Untargeted and targeted lipidomic and metabolomic analysis identified 88 significant, unique and named analytes (Supplementary Datasets 1 and 2) (Fig. S1A–D). Significant alterations were detected in xanthine, acylcarnitines, polyunsaturated fatty acids (PUFAs) and kynurenine (Fig. S1E, F), consistent with previous reports[11,31–35]. Despite differences between the cohorts, no analytes or pathways stood out for explaining the widespread endotheliopathy associated with severe COVID-19. For this reason, we focused on changes detected in plasma proteomes. Label-free proteomics yielded over 100,000 peptide spectral matches, with approximately 8000 unique peptides identified that mapped to 625 unique proteins (Supplementary Dataset 3 and Table S5) (Fig. 1A). Using a cut-off of a P adjusted value <0.1 and an abundance ratio of +/− 1.3, 109 differentially abundant proteins (DAPs) were detected in COVID + versus COVID- patients, with balanced sampling indicating that identified DAPs were not an artifact of severe illness (Fig. 1B). Principle component analysis (PCA) of the identified DAPs resulted in clustering and partial separation of COVID + and COVID- patients (Fig. 1C). Of the 109 identified DAPs, 97 were up-regulated and 12 were down-regulated in COVID + patients compared to COVID- patients (Fig. 1D, G).

We next performed Kyoto Encyclopedia of Genes and Genomes (KEGG) pathway analysis inputting up-regulated DAPs from the COVID + cohort. In line with previous reports[11,36], our analysis demonstrated enrichment in complement and coagulation cascades, neutrophil extracellular trap formation, and extracellular matrix protein (ECM) interactions (Fig. 1E). KEGG analysis also detected alterations in pathways associated with many of the co-morbidities known as risk factors for increased COVID-19 severity, such as diabetes (AGE-RAGE signaling; albeit not significant) and hypertension (renin-angiotensin system). Pathway analysis uncovered differences related to fluid shear stress response and perturbations in glycan homeostasis that have not been reported previously, but which support clinical observations of altered blood flow and RBC hemodynamics as unique aspects of COVID-19 pathogenesis (Fig. 1E). Specific proteins associated with these pathways include various sheddases, enzymes involved in remodeling of the endothelial glycocalyx, and markers of cell stress. Analysis by organ system demonstrated that the most significant alterations were related to the liver, with both up- and down-regulated proteins in COVID + patients mapping to the liver when compared to expression profiling from the human genome atlas (Fig. 1F). This aligns with the induction of a strong hepatic acute phase response to SARS-CoV-2 infection, reflected by significant increases in the presence of all three fibrinogen chains (Fig. 1H) and additionally supported by the most significant KEGG pathway being the complement and coagulation cascades.

### Fibrinogen mediates red blood cell aggregation under static and physiological flow conditions

The significance of the alteration in liver analytes in combination with the unexpected shear stress response pathway in our proteomics analysis was of great interest, especially given our prior observation of increased blood viscosity related to hyperfibrinogenemia in the sickest COVID-19 patients[18]. To examine any link between the hepatic inflammatory mediator fibrinogen and fluid shear stress, a reflection of endothelial injury and dysfunction from altered blood flow, we next performed a series of rheological studies using microvasculature-on-

chip assays that were informed by several clinical observations. Specifically, we noted increased RBC sedimentation in COVID-19 patients strongly correlating with fibrinogen levels (Fig. 2A). We also observed increased RBC sedimentation and marked RBC aggregation under static conditions in blood from COVID-19 patients compared with that from healthy volunteers (Fig. 2B), consistent with findings reported previously[37,38]. To investigate the consequences of this phenomenon

under physiological flow conditions, we tested the influence of fibrinogen on RBC aggregation using advanced microfluidics devices.

Washed RBCs isolated from a healthy blood group O volunteer were combined with increasing concentrations of purified fibrinogen in PBS ranging from 0–900 mg/dL, spanning both the normal range (approximately 200-400 mg/dL) and reported median level in critically ill adult COVID-19 patients (approximately 800 mg/dL)[19].

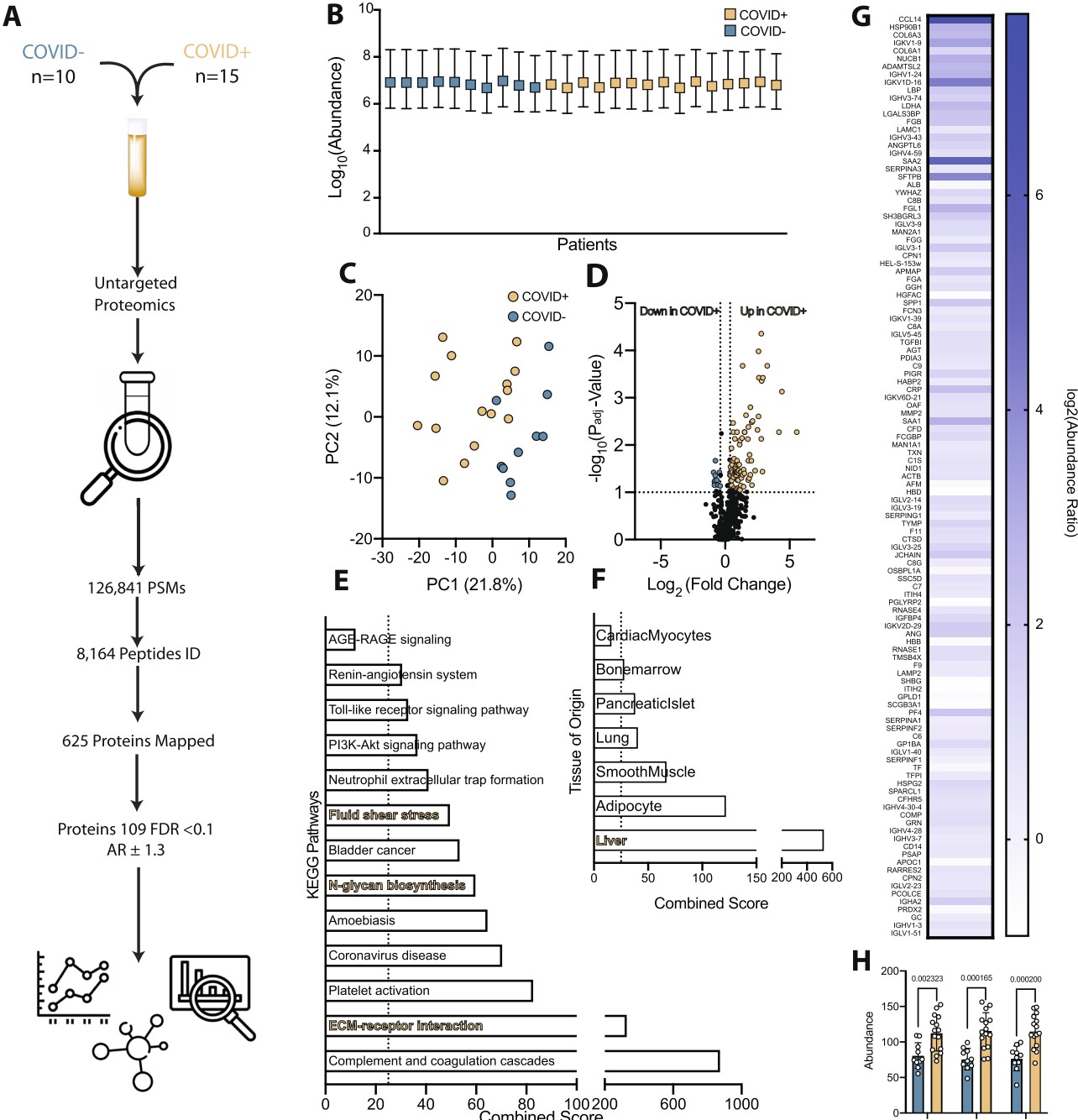

**Fig. 1 | Proteomics analysis comparing adult plasma from critically ill COVID + and COVID- patients uncovers significant alterations in liver-related proteins and hemodynamic-related pathways. A** Schematic overview of the approach and identified species in the adult cohorts of COVID + and COVID- critically ill patients. **B** Relative abundance of detected peptides per patient sample (COVID + n = 15, COVID- n = 10, SD). **C** Principal component analysis (PCA) plot of proteomics data demonstrating clustering and partial separation of plasma samples from COVID + and COVID- adults. **D** Volcano plot showing relative distribution of identified proteins (One-way ANOVA test, Benjamini-Hochberg correction). **E** KEGG pathway analysis of differentially abundant proteins (DAPs) highly

upregulated (Padj > 0.1 AR > 1.4) in COVID + patients highlights alterations in fluid shear stress response pathways (gold) and previously reported changes in ECM receptor interactions (also gold), (Fishers exact test). **F** Comparison of DAPs to the human genome atlas showing relative origin of proteins identified highlights perturbations in liver, lung, and immune cell homeostasis. **G** Heat map of DAPs identified by proteomics analysis that are altered in COVID + adults as compared to COVID- adults. **H** The abundance of all 3 chains of the hepatic acute phase protein fibrinogen is increased in COVID + compared to COVID- adult patients (COVID + n = 15, COVID- n = 10, two-tailed t-test, SD). Source data are provided as a Source Data file.

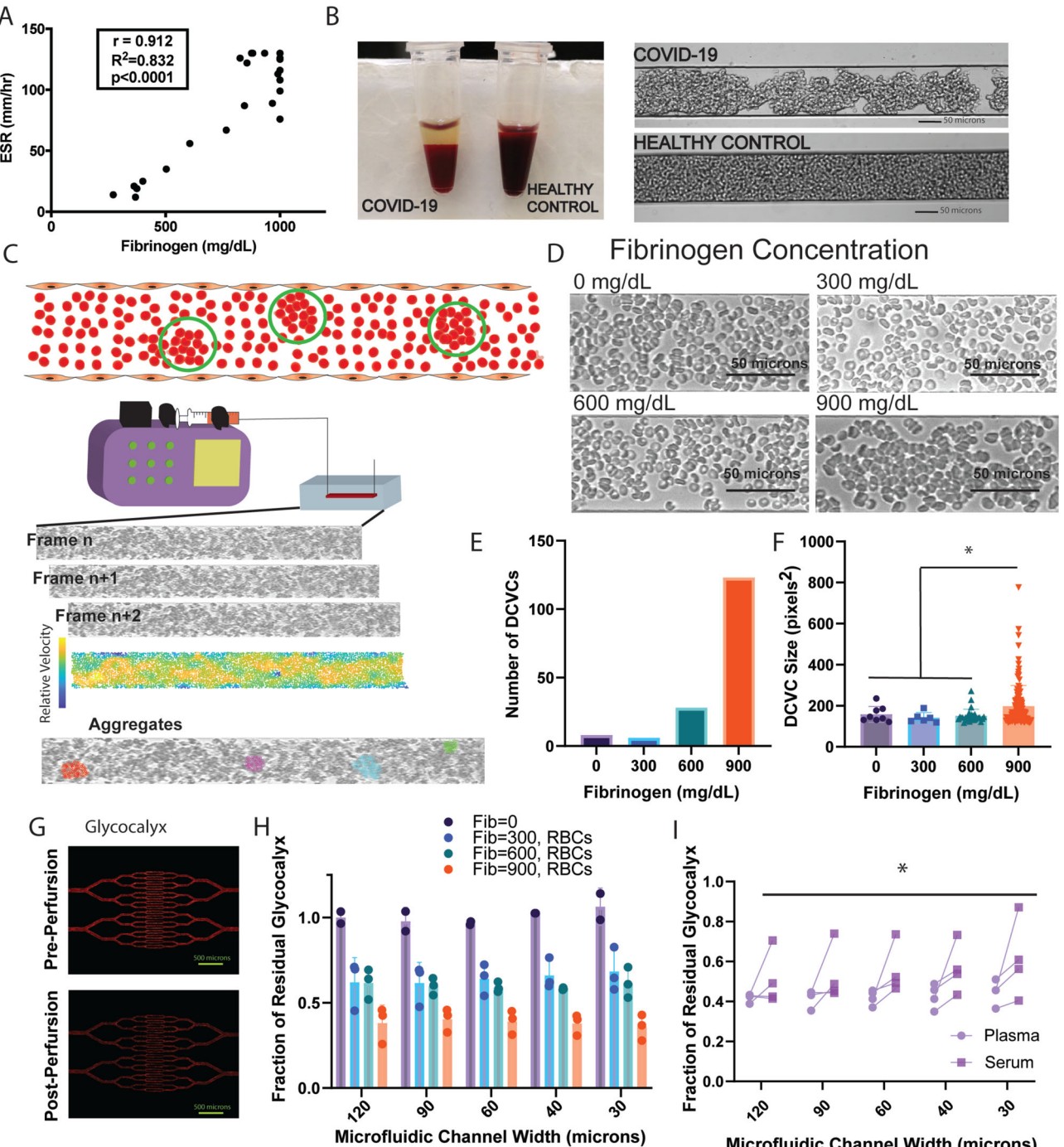

**Fig. 2 | Fibrinogen mediates RBC aggregation and biophysically induces endothelial cell damage. A** Positive correlation between ESR and fibrinogen levels in COVID-19 patients ($p < 0.0001$ via Pearson linear regression). **B** (Left) COVID-19 patient blood shows rapid sedimentation on the benchtop and (Right) aggregation in a microfluidics channel under static conditions compared with healthy control. **C** (Top) Cartoon demonstrating RBC aggregation detected using our velocity field calculations and the differential cell velocity cluster (DCVC) identification. (Bottom) Experimental setup of RBC perfusion into the microfluidics device with a representative time series of frames obtained via video microscopy. Below is a relative velocity map for tracked cell edges with identification of clusters via our criteria for RBC DCVC calculations. **D** Frames from video microscopy at four fibrinogen concentrations showing a qualitative increase in RBC aggregation with increasing fibrinogen. **E** Exponential increase in the number of detected DCVCs with increasing fibrinogen level as compared to minimal amount of aggregation seen at a fibrinogen concentration of 0 mg/dL or at the physiologic level of 300 mg/dL. **F** Increase in the mean DCVC size with increasing fibrinogen concentration. A size of 115 square pixels is approximately 15 RBCs (*indicates a significant difference in DCVC size via unpaired two-tail t-test with Welch's correction, $p < 0.05$; 900 mg/dL vs 0 mg/dL $p = 0.029$, vs 300 mg/dL $p = 0.0010$, vs 600 mg/dL $p < 0.0001$). **G** Endothelial glycocalyx in serially branching microvasculature-on-chip devices stained with fluorescently tagged wheat germ agglutin before (top) and after (bottom) perfusion of RBCs. **H** Average fraction of residual glycocalyx detected in the microfluidics devices from (**G**) at each of the tested fibrinogen levels for $n = 3$ experiments. Significant decrease in residual glycocalyx with increasing fibrinogen level via two-way ANOVA ($p = 0.002$). **I** Measurement of glycocalyx degradation induced by paired plasma and sera (i.e. recalcified fibrinogen-decreased plasma) samples from ($n = 4$) COVID + patients. Pooled results across all channel sizes show a significantly greater degree of glycocalyx degradation in plasma versus sera samples (*$p = 0.0006$ via paired t-test). Unless otherwise indicated, all values are displayed as mean +/- SD. Source data are provided as a Source Data file.

Suspensions of RBCs and fibrinogen were perfused through microfluidics devices (Fig. S2A) approximating the peri-capillary vessel size and associated physiologic flow rate (Fig. 2C). Video microscopy was obtained for each condition for aggregation analysis (Fig. 2D). Differential cell velocity clusters (DCVCs) were used as a measure of RBC aggregation and defined as groups of RBCs traveling at the same relative velocity and in close physical proximity to each other. These DCVCs were quantified from velocity fields using a previously described method which employs a commercially available video tracking algorithm[39] and additional custom software (see Methods as well as Fig. S3 for detailed explanation of image processing workflow). These experiments revealed an increase in the number of RBC aggregates (DCVCs) with increasing fibrinogen levels (Fig. 2E). Additionally, the highest fibrinogen condition showed the largest size and widest size spread of DCVCs (Fig. 2F). DCVCs detected at a fibrinogen concentration of 900 mg/dL were significantly larger than those in any other concentration ($198 \pm 100$ pixels$^2$ vs $158 \pm 36$ pixels$^2$ for 0 mg/dL, vs $141 \pm 23$ pixels$^2$ for 300 mg/dL, and $150 \pm 32$ pixels$^2$ for 600 mg/dL). These results confirm the persistence of fibrinogen-mediated RBC aggregation under physiological flow conditions, and demonstrate that higher fibrinogen concentrations increase both the absolute size and the range of sizes of RBC aggregates.

### Fibrinogen-mediated red blood cell aggregation induces endothelial glycocalyx degradation

We next examined whether fibrinogen-dependent RBC aggregation results in pathologic damage to the endothelium using humanized microfluidics devices. Branched microfluidics devices (Fig. S2B) were cultured with human umbilical vein endothelial cells (HUVECs) for 72 hours and the endothelial glycocalyx layer visualized using a fluorescently tagged wheat germ agglutinin (Fig. 2G). The endothelial glycocalyx was chosen as a target given its important role in regulating endothelial health and its responsiveness to perturbations in the shear stress environment[40]. The fluorescence signal was visualized using a digital microscope and quantified in each isolated channel segment. The same fibrinogen concentrations used for cluster measurement analysis were again combined with healthy volunteer RBCs and perfused into devices for 30 minutes. The devices were then re-imaged and a ratio of residual glycocalyx calculated for each channel segment. These experiments demonstrate a significant reduction in the glycocalyx in the condition with a fibrinogen concentration of 900 mg/dL compared with 600 mg/dL at all channel sizes, and with 300 mg/dL at all but the largest channel size ($p < 0.05$ for all indicated comparisons in the figure). These results indicate that fibrinogen-mediated RBC aggregation directly and pathologically affects the endothelium by promoting glycocalyx degradation.

To investigate if our observations regarding fibrinogen concentration in a tightly controlled in vitro environment translate to clinical samples, we next evaluated the role of fibrinogen in plasma from COVID-19 patients. Plasma was collected from anticoagulated whole blood from four COVID+ patients with abnormally high fibrinogen levels (765 mg/dL to >1000 mg/dL, normal range 200–393 mg/dL). Aliquots were re-calcified and allowed to clot with the supernatant, thereby approximating sera and decreasing the fibrinogen concentration without loss of other non-clotting plasma-based factors (e.g. cytokines, etc). Each of the native samples (Plasma) and re-calcified plasma samples (Serum) were combined with healthy RBCs, perfused into microfluidics devices, and analyzed in the same manner as the fibrinogen concentration experiments. When stratified by channel size, there was no statistically significant change between the residual glycocalyx for plasma versus serum suspensions at any one channel size. However, when all channel sizes were pooled for each individual patient, a paired t-test revealed a significant difference between the two groups (mean difference in residual glycocalyx $0.13 \pm 0.03$, $p = 0.0006$; Fig. 2I). While the difference in residual

glycocalyx between the plasma and serum for any one given patient was not large, this likely reflects variable amounts of residual fibrinogen present in recalcified samples. Nevertheless, these data demonstrate a direct effect of fibrinogen-mediated RBC aggregation contributing to endothelial dysfunction. Fibrinogen has been studied for its role in inducing RBC aggregation previously, yet the phenomenon has been assumed to be a byproduct of systemic inflammation rather than a contributor to acute pathology. These data show a direct pathogenic effect of fibrinogen-mediated RBC aggregation on the endothelium and provide a potential therapeutic target, as suggested by clinical work proposing a survival benefit associated with reduction in fibrinogen levels in COVID-19 patients[30].

### Plasma from patients with COVID-19 induces increased red blood cell aggregation

Building on observations from our in vitro experiments using purified fibrinogen, we next sought to evaluate how plasma from COVID-19 patients influences RBC aggregation compared with critically ill COVID-19-negative patients under biologically relevant conditions. We prospectively collected anticoagulated blood from patients meeting sepsis criteria as a result of SARS-CoV-2 infection (COVID+, $n = 6$) or from another infectious source (non-COVID sepsis, $n = 6$), as well as from healthy volunteers (Healthy, $n = 4$). Patient cohorts were matched for illness severity by SOFA score, age, and number of comorbidities, and samples obtained within 72 h of admission to a single ICU (Table S3). Plasma from each participant was combined with isolated RBCs and aggregation assessed as above. Qualitative evaluation showed increased RBC aggregation in the COVID+ patients as compared to non-COVID sepsis patients or healthy controls (Fig. 3A). When the number of clusters was calculated, plasma from both COVID+ and non-COVID sepsis patients induced a greater number of DCVCs than healthy volunteers (Fig. 3B, $125 \pm 58$ & $101 \pm 57$ vs $17 \pm 7$ respectively, $p < 0.05$). While there was no difference in the number of clusters between COVID+ and non-COVID sepsis, the size of the aggregates in the COVID+ cohort was significantly larger (Fig. 3C, $435 \pm 413$ pixels vs $336 \pm 313$ pixels, $p < 0.0001$).

In addition to aggregate size, we compared the velocity distribution of the DCVCs among the 3 groups. The average velocity of the DCVC was similar between all 3 groups (Fig. 3D, 180 µm/s vs 173 µm/s vs 199 µm/s for COVID+, non-COVID sepsis, and healthy volunteers respectively); however, the variance of pooled aggregate velocities in the COVID+ group was larger than either non-COVID sepsis or healthy controls. To assess this on an individual level, we evaluated the interquartile range (IQR) of aggregate velocities for each participant in the 3 groups (Fig. 3E). The average IQR for the COVID+ cohort was larger than the IQR for the other two groups; however, the difference did not reach significance in comparing the COVID+ vs non-COVID sepsis groups ($57 \pm 48$ µm/sec vs $22 \pm 35$ µm/sec, $p = 0.21$). These findings show that while plasma from both patients with COVID-19 and patients with non-COVID sepsis result in an increase in RBC clustering, the clusters in patients with COVID-19 are larger and exhibit a wider variability in velocity. How larger aggregates in the microvasculature can lead to endothelial cell damage is an intuitive finding, as large RBC aggregates physically degrade endothelial glycocalyx in the microcirculation. However, the variability in velocity is a potentially important finding. Extensive literature including work by our own group demonstrates increased endothelial cell activation and a pro-inflammatory response following exposure to time-variable shear rates as compared to laminar flow;[41] still, this is often in chronic vascular disease and has not been implicated in acute critical illness. The variation in velocity field values, even in a steady state system such as ours, highlights the influence that changes in plasma of patients with COVID-19 can have on rheologic properties in the microvasculature.

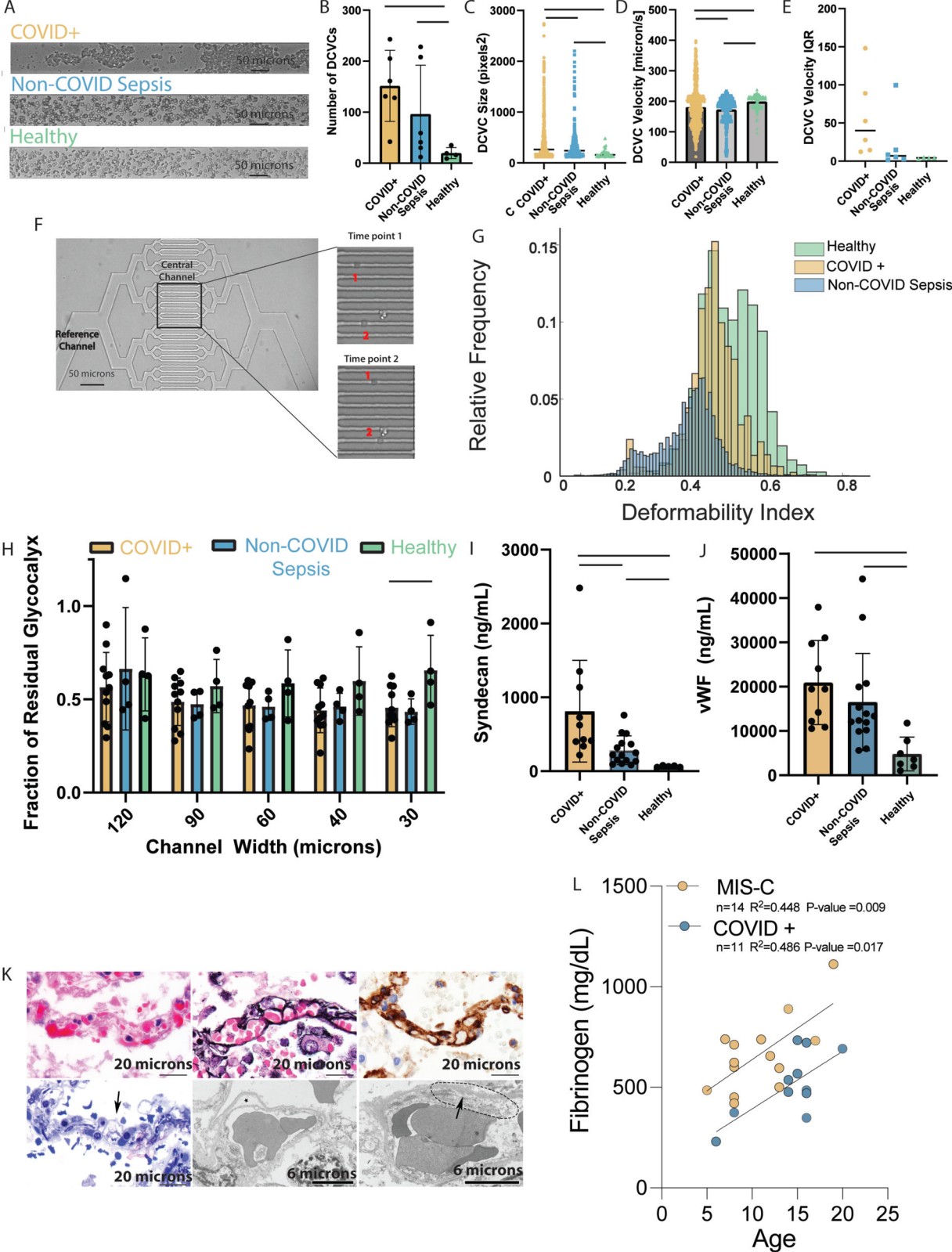

## Differences in RBC membrane deformability between critically ill patients with and without COVID-19

As RBCs enter the capillary bed, membrane deformation is essential to permit cells to pass singly and quickly through the small vessel to maintain consistent flow. Given the evidence of microvascular pathology in patients with COVID-19, we next investigated whether changes in RBC biomechanics at this scale may contribute. To do so,

RBCs were isolated from critically ill patients with COVID-19 (COVID + , $n = 9$), critically ill patients with infectious sources other than SARS-CoV-2 (non-COVID sepsis, $n = 11$) and healthy controls (Healthy, $n = 5$), and washed of all plasma (Table S3). Cells were resuspended in PBS and perfused through microfluidics channels (Fig. S2C). Changes in the single cell RBC deformability indices were evaluated by tracking transit time via video microscopy capture of at least 250 cells per patient or

**Fig. 3 | Plasma from both COVID + and non-COVID sepsis patients results in endothelial glycocalyx damage but from different mechanisms.** Unless otherwise indicated, statistical evaluation was via two tailed t-test, solid bar indicates $p < 0.05$, and data are reported as mean +/− SD. **A** Video microscopy frame showing the RBCs under shear combined with COVID + (top), non-COVID sepsis (middle), or healthy volunteer (bottom) plasma. **B** Increased DCVCs detected with COVID + ($n = 6$) and non-COVID sepsis ($n = 6$) compared with healthy volunteer plasma ($n = 4$, $p = 0.0115$ and $p = 0.0217$). **C** Mean DCVC size in COVID + patients ($n = 743$) was greater compared with non-COVID sepsis ($n = 360$, $p < 0.0001$), and healthy volunteers ($n = 57$, $p = 0.0004$) **D** Average velocity of the DCVCs was similar among all groups; however, there was a significant difference in the variance of pooled measurements by two-tailed t-test with COVID + demonstrating the largest variance. **E** The DCVC velocity IQR of COVID + patients is larger than non-COVID sepsis or healthy controls. **F** (Top left) Two RBCs cross the central channels of deformability assay. **G** Normalized histogram of pooled single-cell deformability indices demonstrating that COVID + RBCs are less deformable than healthy controls, but more than non-COVID sepsis. **H** Plasma from COVID + ($n = 11$, $p = 0.018$) and non-COVID sepsis ($n = 4$, $p = 0.067$) patients induces greater glycocalyx damage in the smallest microfluidics channel compared with healthy volunteers ($n = 4$). **I** COVID + ($n = 11$) patients have increased syndecan-1 versus non-COVID sepsis ($n = 15$, $p = 0.0094$) and healthy volunteers ($n = 5$, $p = 0.0334$); non-COVID sepsis vs heathy ($p = 0.0244$). **J** COVID + and non-COVID sepsis patients have increased levels of vWF compared with healthy controls ($n = 5$, $p = 0.0007$ and $p = 0.0135$ respectively), no significant difference between the two patient populations ($p = 0.316$) **K** Pulmonary capillaries in COVID-19 patients examined post-mortem. Light microscopy (top left), with clogging by erythrocytes and scant fibrin. Jones (top middle) and CD34 (top right) stains show intact basement membrane and endothelial lining. Semi-thin sections for ultrastructural examination (bottom left) show endothelial injury (arrow) characterized by cell swelling and disruption. Transmission electron microscopy (bottom middle, bottom right) shows endothelial cell dehiscence from basement membrane (star) and loss of structural integrity (dotted line) with feathering of the basement membrane (arrow). Representative images from one of 6 COVID-19 decedents analyzed with similar results. **L** Strong correlation (via Pearson correlation coefficient) between maximum fibrinogen level during hospital admission and age in both COVID + pediatric ($p = 0.017$) and MIS-C patients ($p = 0.009$). Source data are provided as a Source Data file.

control (Fig. 3F). Both COVID + and non-COVID sepsis patients exhibited a lower deformability index compared with healthy volunteers, indicating more rigid/less deformable RBCs ($0.40 \pm 0.086$ & $0.33 \pm 0.02$ vs $0.46 \pm 0.088$ respectively) (Fig. 3G). These results indicate that RBCs from patients with COVID-19 are stiffer than those from healthy adults; however, patients with sepsis from causes other than COVID-19 are even less deformable. Stiffened RBCs not only damage the capillary endothelium but also influence the rheologic properties of blood flow, particularly in the microvasculature, as stiffer particles marginate to the vessel wall and flexible cells (like healthy biconcave RBCs) move toward the center. There is precedent in other conditions showing that RBC populations with differential membrane stiffness cause inflammatory endothelial changes[42]. This distribution of RBC membrane stiffness changes the overall cross sectional velocity profile of the microvasculature and consequently the wall shear stress and mechanotransductive signaling at the endothelium. In combination with the aggregation studies, these results suggest that the rheologic disturbances in critical infectious illness differ in patients with SARS-CoV-2 as compared to patients with sepsis from other causes.

### COVID-19 plasma interacts with red blood cells to damage the endothelial glycocalyx in a vessel size-dependent manner

To evaluate if the differences in rheology observed between COVID + , non-COVID sepsis and healthy volunteers results in endothelial damage, we again employed our humanized microvascular microfluidics platform. The same experimental protocol using a microfluidics device cultured with HUVECs was used for COVID + ($n = 11$), non-COVID sepsis ($n = 5$), and healthy controls ($n = 4$) to measure this effect. Previous reports have demonstrated that circulating sheddases are upregulated and can increase the degradation of endothelial glycocalyx[43]. Therefore, to minimize the impact of enzymatic degradation in our model, we designed the assays using a time scale on which biomechanical effects on the glycocalyx could be assessed before any contribution of enzymatic degradation would be expected. Comparing the results using COVID + plasma with that obtained from healthy controls (Fig. 3H), there was a significant increase in glycocalyx degradation in the smallest channel size ($0.44 \pm 0.09$ vs $0.66 \pm 0.16$, $p < 0.05$), but not between the COVID + and non-COVID sepsis cohorts ($0.44 \pm 0.09$ vs $0.43 \pm 0.068$). A repeated measures ANOVA analysis between each of the patient groups and the healthy controls produced a significant difference. These results suggest that changes in circulating plasma components and subsequent biophysical changes can result in endothelial damage in the microvascular system, most pronounced in the smallest caliber vessels, and is a feature of critical illness in both COVID and non-COVID sepsis.

### Indicators of endothelial damage are prominent in patients with COVID-19

Finally, well-established markers of endothelial activation or damage were compared among COVID + ($n = 10$), non-COVID sepsis ($n = 11$), and healthy ($n = 5$) cohorts by enzyme-linked immunoassays (Table S3). COVID + patients had higher levels of syndecan-1, a component of the endothelial glycocalyx, than either non-COVID sepsis patients (Fig. 3I, $812 \pm 65$ ng/mL vs $283 \pm 188$ ng/mL, $p = 0.0094$) or healthy volunteers ($812 \pm 65$ ng/mL vs $65 \pm 10$ ng/mL, $p = 0.03$ and $283 \pm 188$ vs $65 \pm 10$ ng/mL ng/mL, $p = 0.02$). Both COVID + and non-COVID sepsis patients had higher levels of circulating vWF than healthy controls (Fig. 3J) $20,929 \pm 9016$ ng/mL vs $4792 \pm 3513$ ng/mL, $p < 0.001$ and $16518 \pm 10552$ ng/mL vs $4792 \pm 3513$ ng/mL, $p = 0.01$). Moreover, pathologist review of pulmonary microvascular integrity in patients who succumbed to COVID-19 highlighted RBC aggregates clogging the capillaries alongside evidence of endothelial injury (Fig. 3K). Taken together, these clinical findings corroborate microfluidics studies demonstrating the role of endothelial damage from altered blood rheology in COVID-19.

### Multiplatform analysis of plasma from pediatric COVID-19 or MIS-C patients suggests diverging pathophysiology from adult COVID-19

Clinical observations in patients and evidence obtained from animal studies suggest the severity of COVID-19 infection is largely determined by host response and age[44]. To determine if our findings in adults were recapitulated in children, we attempted to perform similar rheological analysis using blood from pediatric patients. Unfortunately, attempts to obtain either fresh plasma prospectively or interpretable data from studies using biobanked pediatric plasma were unsuccessful, largely due to limitations in the microfluidics studies that necessitate large, freshly-collected citrated samples. Nevertheless, clinical data from our cohort of pediatric patients with severe disease demonstrated lower maximum fibrinogen levels during admission in acute COVID ($n = 11$, median = 481 mg/dL, interquartile range 422–630) and MIS-C ($n = 14$, median = 640 mg/dL, interquartile 524–736) (Table S4), compared to published reports in adults (794 mg/dL, interquartile range 583–933[19], as levels from much of our adult cohort were unavailable). Interestingly, fibrinogen levels in these pediatric patients demonstrated a positive correlation with age (Fig. 3L) (Table S4). Taken together with the fibrinogen-concentration studies (Fig. 2), these data suggest fibrinogen-mediated RBC aggregation is not likely to occur in pediatric patients with acute COVID given the lower degree of hyperfibrinogenemia.

We thus turned back to multiomics analysis to gain insight into alternative mechanisms characterizing SARS-CoV-2 pathogenesis in

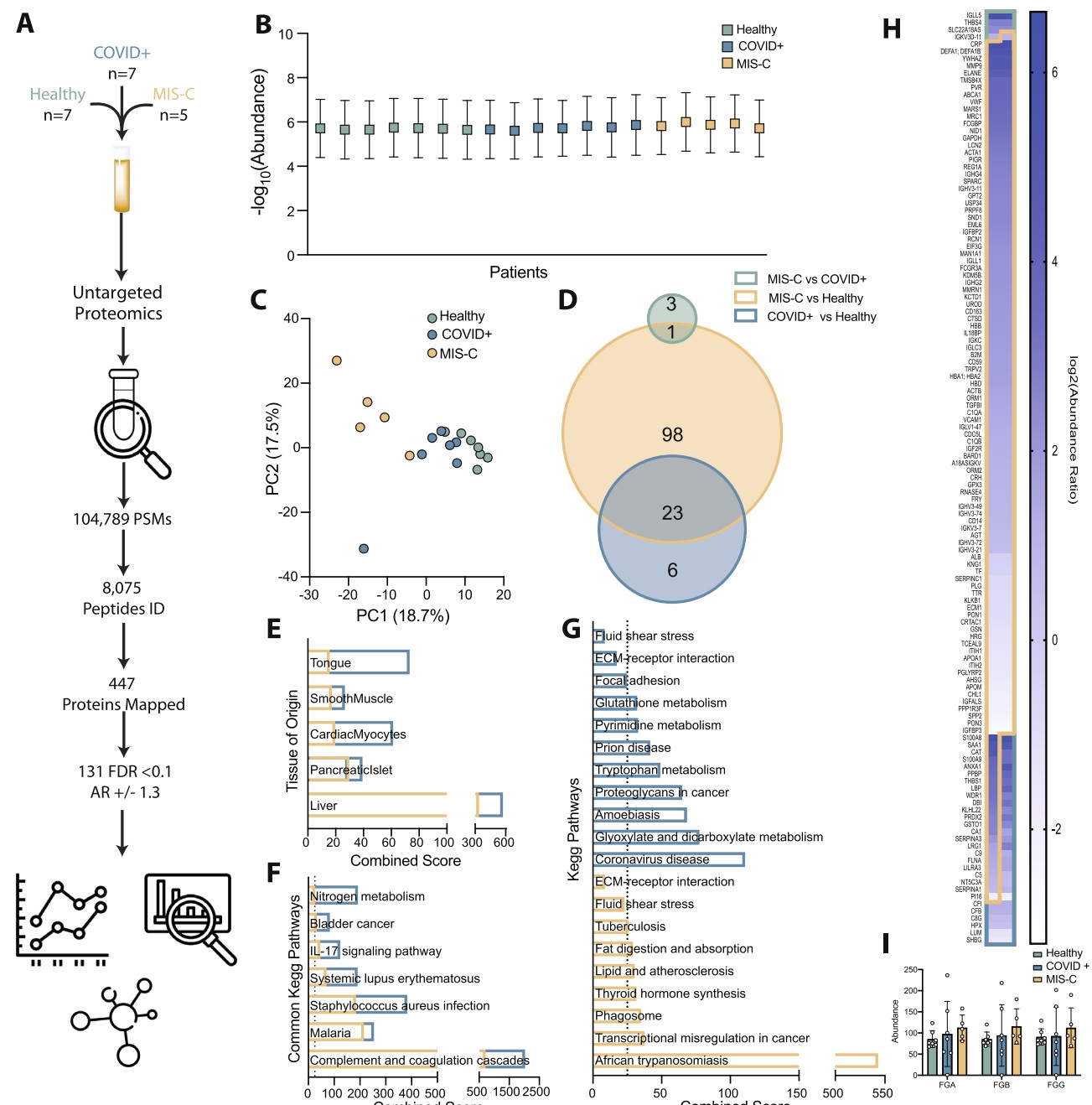

**Fig. 4 | Proteomics analysis of pediatric plasma from patients with acute COVID-19 or post-infectious MIS-C. A** Overview of untargeted plasma proteomics on sera from pediatric patients with COVID-19 (COVID +), MIS-C, or healthy controls. **B** Relative abundance of detected peptides per sample (Healthy $n = 7$, COVID + $n = 7$, MIS-C $n = 5$, SD). **C** PCA plot of proteomics data for pediatric group. **D** Venn diagram showing the relative distribution of differentially abundant proteins (DAPs). **E** Comparison of DAPs to the human genome atlas highlights perturbations in liver and lung; gold reflects MIS-C compared to healthy controls while blue reflects COVID + compared to healthy controls. **F** KEGG pathway analysis

highlights alterations in pathways common between MIS-C and COVID + cohorts. **G** KEGG pathway analysis of those significant in either MIS-C (gold) or COVID + (blue) highlights perturbations unique to each group, neither of which achieve significance for the fluid shear stress or ECM-receptor interaction pathways identified in the adult cohort. **H** Heat map of identified DAPs altered upon SARS-CoV-2 infection in children. **I** The abundance of all 3 chains of the hepatic acute phase protein fibrinogen is not significantly increased in pediatric COVID + or MIS-C patients (Healthy $n = 7$, COVID + $n = 7$, MIS-C $n = 5$, two-way ANOVA, SD). Source data are provided as a Source Data file.

children. Plasma samples were obtained from healthy children ($n = 7$), children admitted with PCR-confirmed COVID-19 (COVID +, $n = 7$), and children admitted with a diagnosis of MIS-C ($n = 5$) (Table S5). Label-free proteomics of plasma detected over 100,000 peptide spectrum matches covering 447 proteins, with 131 being statistically significant between all three groups (Supplementary Dataset 4) (Fig. 4A and Fig. S4A). Balanced sampling indicated that identified DAPs were not an artifact (Fig. 4B). PCA analysis demonstrated that PC1 best

differentiates healthy versus COVID + pediatric patients, with greater separation between children with COVID-19 and those with MIS-C (Fig. 4C). To assess unique drivers of COVID versus MIS-C, we performed pairwise comparisons between each of our three cohorts, finding that MIS-C has the most significant perturbations (Fig. 4D, G). Interestingly, our analysis showed minimal significant alterations between COVID + children and MIS-C children, as they share ~80% of identified DAPs when compared to healthy controls. In general, shared

DAPs were more significantly altered in MIS-C, suggesting that MIS-C results in exacerbation of the changes occurring upon acute COVID-19 infection in children.

Pathway analysis comparing DAPs in our pediatric cohort to the human genome atlas detected similar tissues of origin between pediatric COVID-19 or MIS-C that we observed in adult patients (Fig. 4E). Specifically, KEGG pathway analysis combining the DAPs that were increased or decreased for each pair wise comparison uncovered common changes related to complement and coagulation cascades, inflammatory processes (such as IL-17A signaling, various infections, and even cancer) and autoimmune disease (Fig. 4F). This same analysis also uncovered changes unique to acute COVID + children and MIS-C. Acute COVID + children showed alterations in various infectious processes, including coronavirus infection, and metabolic processes associated with nucleotides and proteins. MIS-C showed similarity to African trypanosomiasis and alterations in metabolic processes associated with lipid metabolism (Fig. 4G). Interestingly, children with MIS-C but not children with acute COVID demonstrated some alterations in fluid shear stress response pathway proteins, though this did not reach our stringent cut-off for statistical significance. Considering our microfluidics data, this may reflect the higher fibrinogen levels in the MIS-C versus acute pediatric COVID cohort; however, other non-fibrinogen mediators may cause the alterations highlighting this pathway, given the fact that MIS-C is characterized by a highly inflammatory vasculitis. Proteomics supports this supposition, as no significant increase in the abundance of fibrinogen chains was detected in acute COVID + or MIS-C children (Fig. 4I).

Notably, KEGG pathway analysis detected changes in our pediatric cohort not present in COVID + adults, mainly related to various metabolic processes, as well as those distinguishing acute pediatric COVID from post-infectious MIS-C. To delineate these unique aspects of SARS-CoV-2 in children, we compared DAPs between the pediatric COVID + and adult COVID + cohorts. Only 28 DAPs were shared, with proteins involved in the complement and coagulation cascades, alterations in the extracellular matrix receptor interactions, and indicators of liver injury. Our analysis also detected proteins that trended with pediatric disease severity (Fig. S4B, C), as well as those involved in pathways distinguishing pediatric COVID-19 and MIS-C (Fig. 4G). Proteins that increased as a function of disease severity included those associated with coagulation, inflammation, and immune cell function, whereas proteins that were decreased tended to be associated with glycoproteins, inhibitors of various proteolytic processes, and lipoprotein metabolism.

Metabolites and lipids are cellular effectors which reflect host cell biological processes, and alterations from their normal levels can inform the pathophysiology underlying a particular disease state. This consideration, combined with alterations in metabolism as suggested by our proteomic analysis, led us to perform global high-resolution metabolomics and lipidomics on our pediatric cohorts. Analysis identified over 2000 features, with 206 being statistically significant between the groups (Fig. 5A, Supplementary Datasets 5 and 6). Unbiased hierarchal clustering using all identified analytes shows partial separation between each group, with COVID + and MIS-C clustering together (Fig. 5B). Similar to our proteomic analysis, most of these differentially abundant analytes (DAAs) are driven by comparison between MIS-C and healthy controls, with less significant changes detected between other pairwise comparisons (Fig. 5C). In agreement with this analysis, we also observed separation in our PCA, with PC1 driving most differences between disease states (Fig. 5D). Unlike the adult cohort, variables most responsible for the separation were predominantly lipids, many of which contained polyunsaturated fatty acyl chains. While we cannot identify the precise location of the desaturation on each fatty acyl chain, this analysis points to alterations in essential fatty acid homeostasis, which is in line with detected changes in nutrient uptake and changes in systemic inflammation associated

with MIS-C (Fig. 5E). Looking at the most significantly altered metabolites when comparing healthy controls and either COVID + children or MIS-C further supports this, as alterations in inflammatory mediators derived from either omega 3 or omega 6 fatty acids predominate (Fig. 5F). KEGG pathway analysis comparing each cohort within the pediatric data set shows expected areas of overlap as well as unique differences (Fig. 5G). Increases in metabolites such lactic acid and inflammatory mediators (20-HETE and 9-HpODE) seemed to be diagnostic for MIS-C, whereas analytes such as DHA and aconitic acid were significantly decreased.

Finally, to gain additional insight into the underlying mechanisms driving the differential presentation of COVID-19 in adults versus children, we measured levels of COVID-19-associated cytokines in our adult and pediatric multiomics cohorts (Tables S1 and 5) and correlated these with our multiomics data. Both COVID + children and children with MIS-C present with a different cytokine profile as compared to COVID + adults (Fig. 6A). Overall, our results demonstrate similar alterations as those previously reported[45]. Given the well-defined functional roles of these cytokines in combination with the differentiating capability of our metabolomics data, we performed an integrated correlation analysis between lipid, metabolite, and cytokine levels. Adult and pediatric acute COVID cohorts demonstrated less correlation between cytokines and DAAs as compared to the MIS-C cohort (Fig. 6B). Adults tended to have strong correlations with IL-6, while COVID + children seemed to have a strong correlation with IL-12p70. Notably, IL-6 is known to be associated with increases in fibrinogen, including in patients with COVID-19[19]. This differs drastically from MIS-C, where we saw an increased correlation between all cytokines and analytes, which may reflect an increased metabolic burden associated with immune cell activation.

Network analysis of the integrative correlation data demonstrated positive (red lines) and negative (blue lines) correlations in each cohort (Fig. 6C, D, E). Communities, shown by assorted colors, were determined by the number of connections between measured cytokines (depicted as squares) and DAAs (depicted as circles). This analysis shows less connectivity between cytokines and metabolites in the adult cohort (Fig. 6C), compared to acute COVID infection in children (Fig. 6D). Children with MIS-C demonstrated a greatly magnified effect (Fig. 6E), further supporting the notion that a heightened inflammatory state drives much of the metabolic differences observed in MIS-C. In general, our data suggests that IL-10 correlates strongly with IFN-γ during COVID infection, with each other assayed cytokine displaying a more distinct set of correlated analytes. Network analysis also suggests a difference in the effect of elevated cytokines in adults as compared to pediatrics patients, as adults display distinct communities that correlate well to specific analytes. Finally, our analysis shows a muddling of these networks in various disease states, with 5 cytokine-analyte communities present in the adult COVID + cohort, 3 present in the pediatric COVID + cohort, and 1 in the MIS-C cohort.

## Discussion

While the scientific and medical communities have made considerable strides in mitigating SARS-CoV-2 infection through vaccines and antiviral therapies, there remains a fundamental lack of knowledge in understanding the molecular mechanisms responsible for severe COVID-19. Both pediatric and adult COVID-19 patients demonstrate a wide spectrum of disease severity, and critically ill patients remain difficult to treat and even refractory to standard treatment - in large part because the pathophysiologic mechanisms underlying the development of severe COVID-19 have not been defined. In this study we used a comprehensive set of complementary technologies - coupling multiomics biochemical studies with sophisticated microvasculature-on-chip assays - to uncover a mechanism of systemic pathogenesis in adult patients stemming from biophysically-induced microvascular damage. This mechanism of COVID-19 pathogenesis in

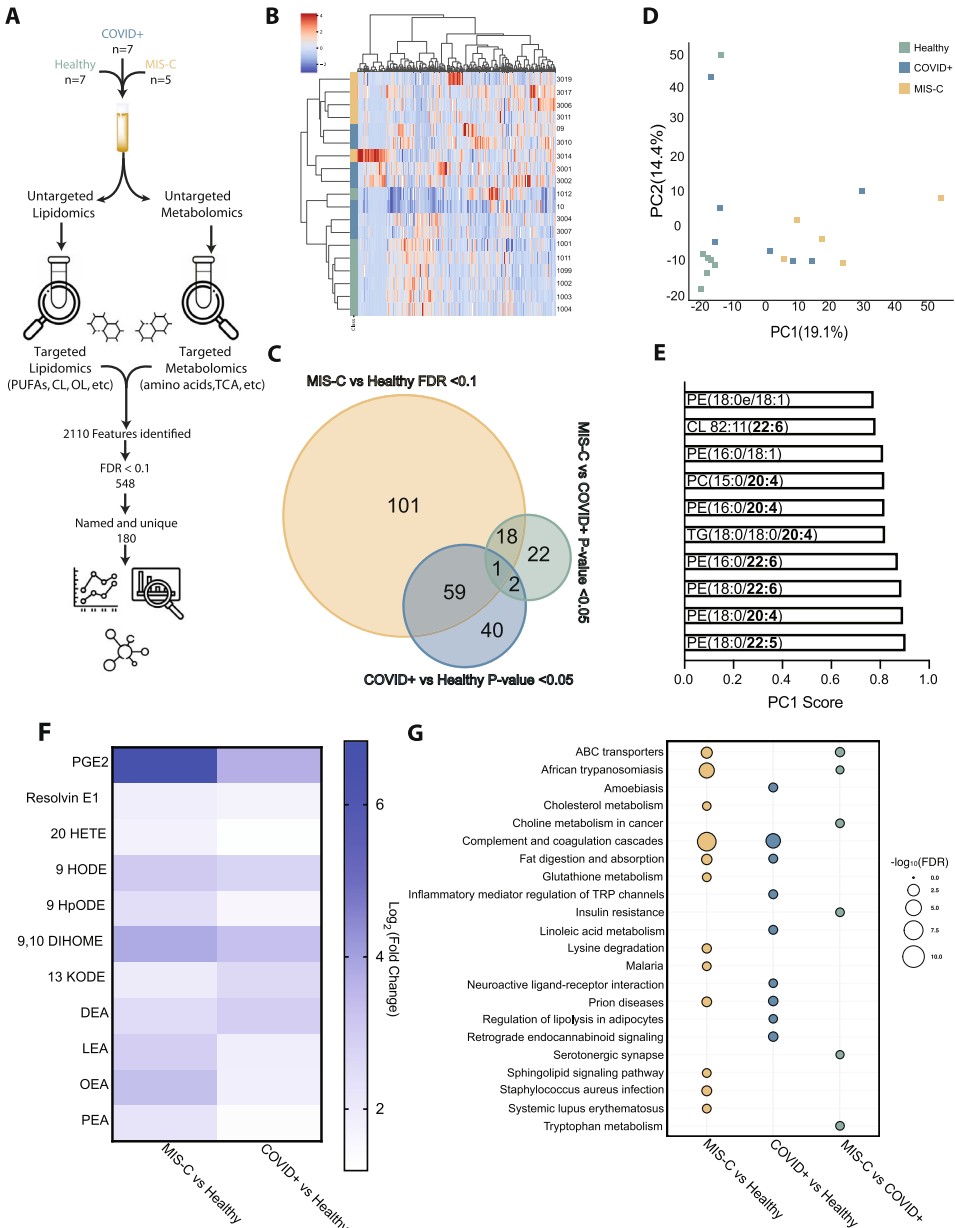

**Fig. 5 | Lipidomics and metabolomics analysis of pediatric cohorts reveal alterations in nutrient absorption and metabolism and increases in inflammatory mediators. A** Workflow showing targeted and untargeted assays used to interrogate lipid and metabolite changes associated with acute pediatric COVID + and MIS-C. **B** Unbiased hierarchal clustering using all confidently annotated analytes. **C** Overlap between each pairwise comparison with respective statistical criterion. **D** Principal component analysis (PCA) using all confidently annotated analytes. **E** Top 10 analytes that define separation of principal component one. **F** Log 2 fold change of significant features present in both COVID + and MIS-C (p-adjusted value <0.1) show changes in inflammatory mediators (Limma, Benjamini-Hochberg correction). **G** Bubble Plot of KEGG pathway analysis comparing groups highlights unique aspects of MIS-C using analytes with a *p*-value <0.05 and DAPs determined from proteomic analysis (Fisher Exact test). Source data are provided as a Source Data file.

adults appears distinct from the immune dysregulation characterizing acute pediatric COVID-19 or post-infectious MIS-C (Fig. 7).

Our proteomic analysis in adult patients confirms associations of COVID-19 with previously described changes in host response pathways, including the complement and coagulation cascades, NET formation, and ECM interactions[11,36]. These data support laboratory findings exploring these pathways and highlight the convergence of COVID-19-induced pathophysiology on endothelial cell health and blood-endothelial interactions[46–49]. Compared with previous reports, however, our proteomic analysis also uncovered unappreciated changes related to fluid shear stress responsive pathways, including alterations in VCAM1 (important in recruiting circulating inflammatory mediators to the endothelial surface), GSTO1 and TXN (cysteine-based redox regulatory and signaling proteins), MMP2 (related to extracellular matrix homeostasis), and ACTB (beta actin, part of the intracellular mechanotransductive pathway). Perturbations in shear stress profiles have long been considered indicators of an unhealthy endothelium, as well as implicated in the development of chronic vascular diseases[50–53]. Thus, our findings provide mechanistic insight into why epidemiological studies demonstrate worse clinical outcomes in COVID-19 patients with certain risk factors, such as those with diabetes, hypertension, underlying vasculopathy or advanced age, all of which are associated with compromised endothelial function.

Microfluidics-based studies in combination with image-capture algorithms allowed for the specific elucidation of COVID-19-associated changes in blood biomechanical properties at the microvascular

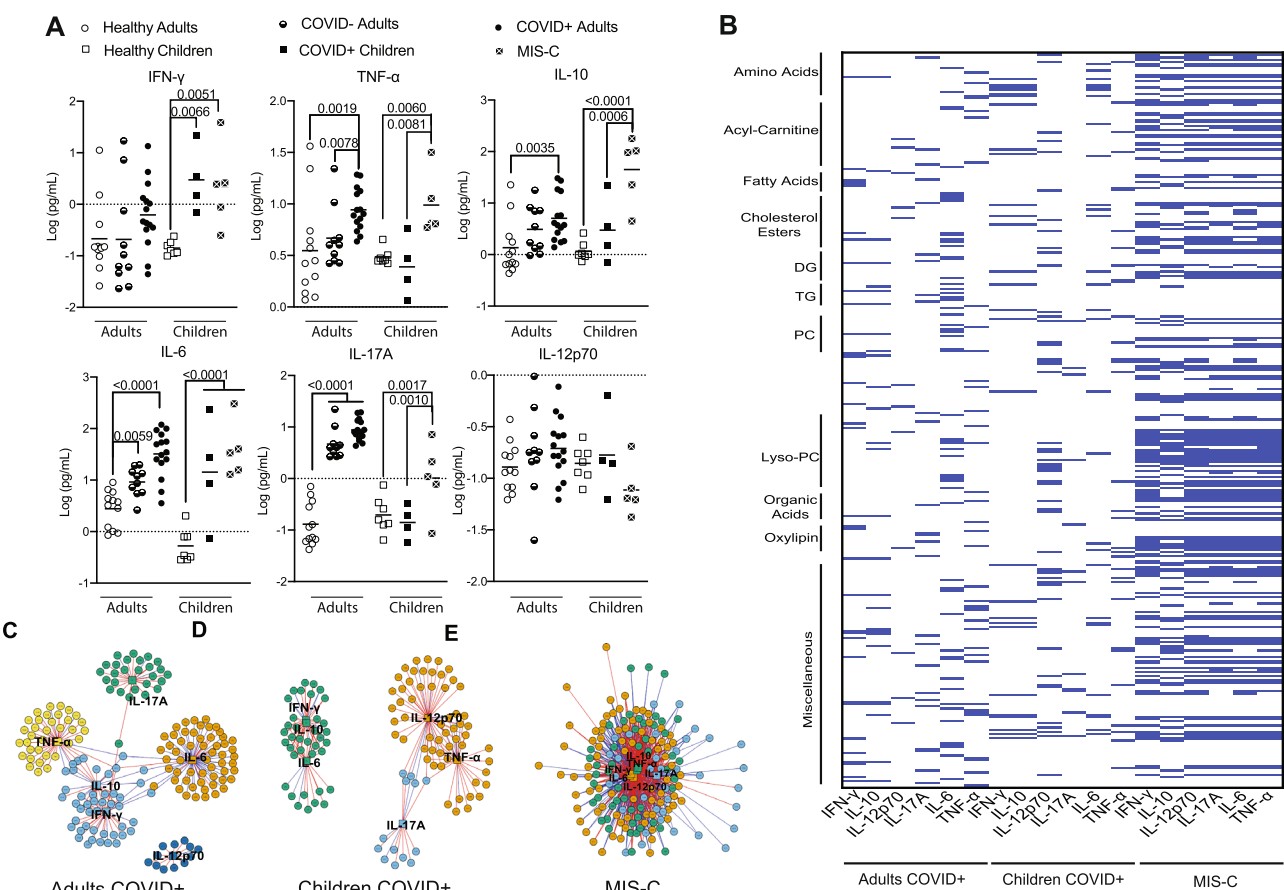

**Fig. 6 | Comparison of cytokine responses in COVID + adults, COVID + children or children with MIS-C. A** Plasma levels of cytokines present in various cohorts indicated. Both COVID + children (n = 4) and children with MIS-C (n = 5) present with a different cytokine profile compared to healthy children (n = 7). COVID + adults (n = 15) have the highest increase in these cytokines followed by severely ill COVID− adults (n = 10) and healthy controls (n = 12), (one-way ANOVA, p-value shown above). **B** Comparison of cytokines with analytes identified by multiomics studies further demonstrates distinct pathogenesis of COVID-19 in adults as compared to pediatric patients (Supplementary Datasets 7 and 8). **C**–**E** Network analysis of the metabolites that correlate with cytokines in COVID + adults (**C**), COVID + children (**D**) or children with MIS-C (**E**). Network plots were generated using a centrality cut-off determined by xmWAS[97]. Information for these plots is provided in the Supplementary Information (Supplementary Datasets 7–10). Source data are provided as a Source Data file.

scale[54], including the influence of altered rheology from pathologically elevated fibrinogen on human endothelium. RBC properties contributing to microvascular changes in critical infective illness have been reported previously and represent an active and growing area of research[55–57]. Our work now demonstrates that plasma from adult COVID-19 patients causes significant RBC aggregation under flow, more so than non-COVID sepsis, and that fibrinogen-mediated aggregation directly damages the endothelial glycocalyx. Disturbances in RBC biophysical properties – membrane shape and single-cell flow behavior – have likewise been reported in recent literature and have been shown to occur following exposure to COVID-19 patient plasma[58]. Notably, here we find that COVID-19 induces RBC aggregation to such a pathologic degree that RBC clusters are not only stable under physiologic flow conditions but also stable enough to cause microvascular injury, which is distinct from weaker electrostatic forces that may cause rouleaux formation of RBC under static conditions.

While we highlight the influence of fibrinogen on RBC behavior in COVID-19 and its effect on the endothelial glycocalyx, the contribution of additional pathways, including other circulating proteins promoting RBC aggregation or changes to RBC membranes at either the phospholipid or membrane protein level, remain to be explored. From our findings, however, we posit that this cascade contributes to the impaired microvascular perfusion, thrombosis, and endotheliopathy observed clinically and in post-mortem studies[59–62]. Importantly, the 30-minute time scale on which these experiments were conducted

suggests that the destruction of the protective glycocalyx layer is not a result of enzymatic degradation, but that direct alterations in the shear stress profiles contribute to endothelial pathology. Moreover, changes identified in the plasma proteome demonstrate alterations in glycan homeostasis related to endothelial glycocalyx maintenance, which have not yet been reported but are in agreement with prior studies[63–65]. The importance of these findings is two-fold. First, the ability of RBC aggregation to directly damage the endothelial glycocalyx has not been reported previously and presents a paradigm for understanding endotheliopathy in critical illness. Second, while patients with critical illness from either COVID-19 or non-COVID sepsis demonstrate microvascular injury, the underlying mechanisms are distinct and provide insight into the unique pathophysiology associated with SARS-CoV-2 infection.

Our metabolomic and lipidomic analysis of adult and pediatric COVID-19 plasma similarly agree with previously reported findings, including changes in our adult cohort related to xanthine, acylcarnitines, PUFAs, and lactic acid[66–69]. These metabolites and lipid by-products have been implicated previously in inflammatory processes[70], and, more recently, others have reported a positive correlation between certain PUFAs and increasing BMI or age in COVID-19 patients[71]. Previous lipidomic studies of COVID-19 patients have also identified alterations in lipids associated with RBC membrane constituents, again consistent with our data[72]. These findings suggest a role for changes in RBC membranes, including stiffness or

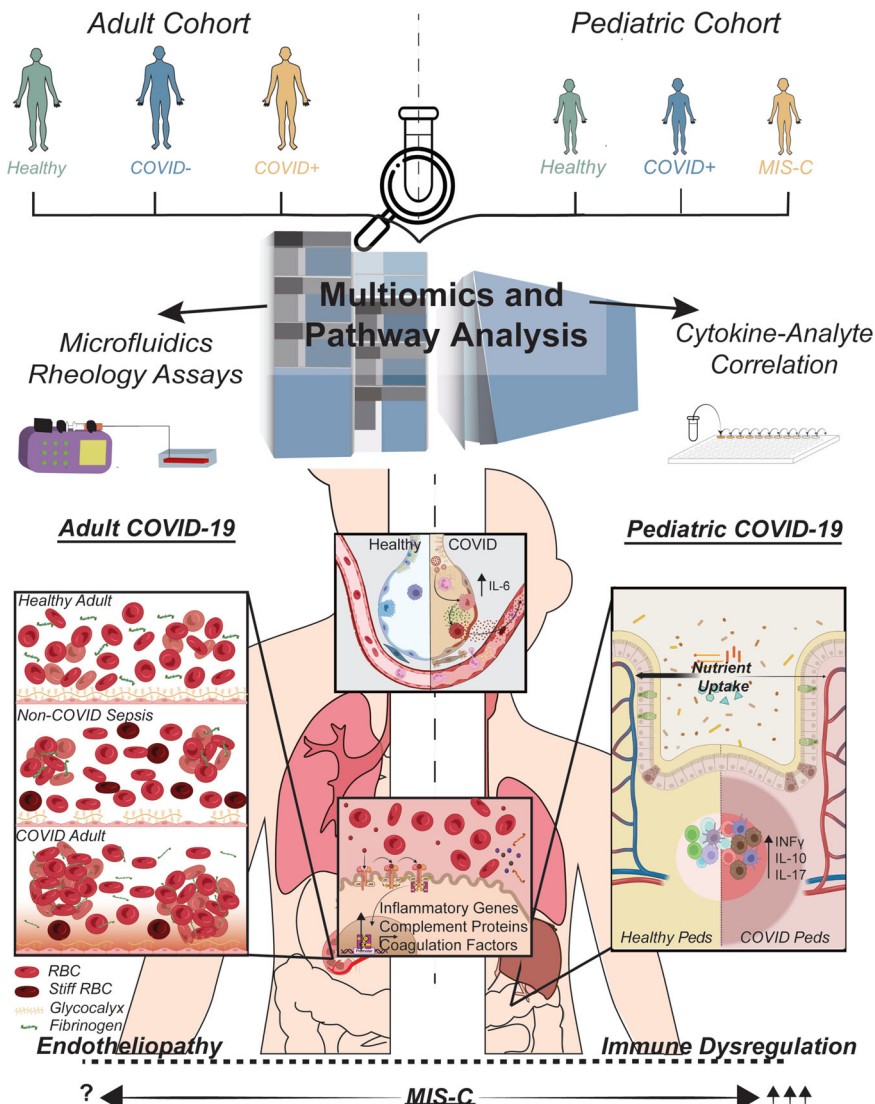

**Fig. 7 | SARS-CoV-2 infection induces distinct changes in adult versus pediatric patients.** Although SARS-CoV-2 is a respiratory virus, multiomics analysis demonstrates systemic changes related to inflammation and the hepatic acute phase response. In adults, the lung-liver axis alterations associated with COVID-19 lead to hypercoagulability and alterations in hemodynamics. As demonstrated in our microfluidics assays, pathologically elevated fibrinogen causes RBC aggregation that damages the endothelial glycocalyx, resulting in endotheliopathy and increased potential for microvascular thrombi formation. This appears to be a distinct mechanism in patients with COVID-19 when compared with patients with sepsis from other infectious causes. In contrast, analysis of samples from pediatric patients suggests that alterations in inflammatory signals cause immune dysregulation. MIS-C demonstrates greater immune dysregulation than acute pediatric COVID and results in a hyper-inflamed state, with an uncertain contribution of fibrinogen in mediating the vasculopathy observed clinically in MIS-C.

deformability, which are directly supported by our rheological assays, although it is worth noting that the changes in RBC rigidity were significantly less in COVID-19 patients than in our non-COVID sepsis cohort. Finally, targeted combinatorial analysis revealed convergence of these pathways on the liver[73], providing an important link between COVID-induced activation of the hepatic acute phase response, including fibrinogen production, and changes in blood rheology at the microvascular level.

Given the different clinical phenotype of SARS-CoV-2 infection in adults and children, as well as the spectrum of disease seen in both populations, we performed comparative analysis on plasma from children with COVID-19 or with MIS-C. The profiles of pediatric patients with COVID-19 or with MIS-C demonstrate increases in proteins involved in pro-inflammatory pathways, including complement and coagulation cascade alterations, similarly to what was seen in the adult COVID-19 population (Fig. S4D and Supplementary Dataset 11). Unique pathways identified in children, compared with

adult COVID-19, include those related to parasitic infection and nutrient digestion and absorption. Additionally, and in contrast to adult patients, elevations in cytokine levels demonstrated strong correlation with disease severity in pediatric patients. This is consistent with cytokine profiling studies previously reported in pediatric patients and our current understanding of the role of immune dysregulation in driving systemic disease in children[13,74–76]. Notably, the vascular shear stress response pathway was not significant in pediatric acute COVID or MIS-C, though it was near to achieving statistical significance in MIS-C. Finally, in comparing the multiomics profiles of children with COVID versus those with MIS-C, dramatic decreases in aconitic acid and IL-17A were seen in MIS-C. IL-17 has been shown to correlate with disease severity in other acute systemic vascular diseases, like Kawasaki Disease, and is implicated in several autoimmune conditions. Together, these data support the idea that a hyperinflammatory process contributes to the development of MIS-C in pediatric patients[77–80].

Several significant limitations warrant further consideration. First, our cohorts for multiomics analysis are small, especially in the pediatric groups, and are not well-matched for certain demographic characteristics known to influence multiomics profiles. Importantly, analysis of our adult cohort comparing only critically ill patients with or without COVID would not detect changes in other pathways that may be shared among disease states, such as sepsis or acute respiratory distress syndrome. In addition, clinical information related to cell differentials, including assessing lymphocyte counts for the presence of significant lymphopenia, which is a prominent feature of SARS-CoV-2 infection and associated with severe disease[81], was unavailable. The effect of cellular composition to overall plasma profiles is uncertain, but anticipated to be contributory, and thus requires additional study. Nevertheless, advantages of our limited cohort collected very early in the pandemic include the near absence of immunomodulatory drugs that have since become standard, given their clinical efficacy, as well as the absence of confounding variables such as patient vaccination status or viral strain.

Additional study limitations include the lack of targeting additional moieties suggested by our omics pathway analysis or detected analytes. In addition, we did not employ targeted LC/MS/MS for known COVID-19-associated metabolites and proteins highlighted in our omics pathway analysis. For example, observed differences in nutrient uptake may reflect intestinal inflammation or even leaky gut syndrome in our pediatric cohort, which has been reported in MIS-C, and could be corroborated by identifying LPS or zonulin in a targeted approach[82]. Finally, our microfluidics-based assays were unable to be performed in the pediatric cohort due to a paucity of severely ill patients available for prospective enrollment and the unsuitability of biobanked samples for these sensitive studies. Instead, we relied on clinical measures of fibrinogen coupled with fibrinogen dose-escalation studies and our multiomics data to infer that the rheological alterations described here for acute COVID are unique to adults. Future studies using larger patient cohorts, including pediatric patients with MIS-C or acute pediatric patients with significant hyperfibrinogenemia, are necessary to establish the role of altered blood rheology as a contributor to disease morbidity and to guide any clinical application, including interventions aimed at decreasing fibrinogen and/or normalizing blood rheology. In addition, studies investigating whether the predominance of alterations in hepatic analytes reflects a generalized inflammatory response induced by SARS-CoV-2 versus a more direct effect of the virus on the liver are warranted.

While our study evaluated patients in the acute stages of illness, patients with the more recently recognized clinical phenomenon of long COVID likely represent a relevant population for future comparison, specifically to investigate whether ongoing alteration in plasma components or blood rheology parameters play a role in the persistence of long COVID symptoms[83,84]. In addition, examining the role of altered blood rheology in driving endothelial damage in COVID-19 may turn out to be relevant in certain children, such as those with underlying vascular disease or those in late teenage years. Nevertheless, our data gives insight into potential approaches for patient management protocols and treatment strategies in severely ill adults, including the use of interventions such as therapeutic plasma exchange or fibrinolytic therapies to remove pathologically elevated fibrinogen, restore normal microvascular perfusion, and thereby mitigate systemic organ dysfunction.

## Methods

### Patients and patient samples

All studies were conducted according to the guidelines of the Declaration of Helsinki and approved by the Institutional Review Board (IRB, or Ethics Committee) of Emory University (IRB00000723, IRB00087446, STUDY00000401, and STUDY00000510). All human samples were obtained following informed consent in compliance with Emory's IRB determination. Patients consenting to participation in this study were not compensated. Gender was determined based on clinical chart review and was not specifically considered in the study design. Demographic information for all patients was obtained by electronic chart review and is provided in the Supplementary Information (Tables S1, S3, S4, and S5).

Adult plasma samples used in proteomics, lipidomics, and metabolomics studies were obtained from residual clinical specimens available from patient testing at Emory University Hospital. Clinical samples were collected by venipuncture into EDTA-containing tubes, separated from cells within 4 hours of phlebotomy, used in clinical testing, and then held at 4 °C for up to 24 hours. Residual specimens were then collected, aliquoted, and frozen at −80 °C until use in multiomics studies. Plasma for healthy controls was obtained from healthy adult volunteer donors ≥18 years old meeting AABB eligibility criteria for source plasma donation. For microfluidics studies, patients were identified within 48 hours of hospital admission and screened for a diagnosis of sepsis by Sepsis-III criteria as a result of PCR positive SARS-CoV-2[85]. Patients or their legal next of kin were contacted for consent according to the approved IRB protocol. Blood was collected in citrated vacutainer collection tube (BD) for same-day experimentation and plasma was isolated and frozen at −80 °C for later use.

Pediatric plasma samples used in proteomics, lipidomics, and metabolomics studies were collected from patients 0 to 21 years of age hospitalized at Children's Healthcare of Atlanta. Prospectively collected blood and/or residual samples were collected following informed consent and assent, as appropriate for age. Children were classified as having COVID-19 if they were hospitalized with a clinically compatible illness and positive SARS-CoV-2 testing by nasopharyngeal RT-PCR. Children were classified as having MIS-C if they met the CDC case definition[86]: fever, laboratory evidence of inflammation, evidence of more than 2 organs affected, plus evidence of SARS-CoV-2 infection (current or recent nasopharyngeal RT-PCR–positive result for SARS-CoV-2 or positive SARS-CoV-2 serology) or exposure to an individual with confirmed or suspected COVID-19 within the preceding 4 weeks. Demographic, clinical, and outcome data were abstracted from the electronic medical record and entered into a Research Electronic Data Capture (REDCap) database hosted at Emory University[87,88]. The controls were healthy outpatient children who participated in a healthy phlebotomy study.

### Proteomics

**Preparation of plasma samples for proteomics.** Patient samples were de-identified prior to receipt and subsequent experimentation and analysis performed blinded. All reagents were of HPLC grade or higher. Plasma samples were thawed on ice and gently mixed by agitating the side of the tube. The 10 μL of neat plasma was diluted 1:10 in 100 mM triethylammonium biocarbonate (TEAB, Sigma Aldrich). Protein concentration was then determined by performing a Bradford protein assay. 150 μg of total protein was transferred to a low binding centrifuge tube and diluted with sodium deoxycholate to a final concentration of 1% (w/v) and 5 mM tris(2-carboxyethyl)phosphine (TCEP, Thermofisher). The sample was then incubated at 60 °C for 30 minutes. After cooling for 5 min at room temperature, alkylation of the cysteines was initiated by the addition of iodoacetamide (SigmaAldrich) to a final concentration of 10 mM. This solution was incubated in the dark for 30 minutes at 37 °C. The alkylation reaction was quenched by the addition of DL-dithiothreitol (DTT, Sigma Aldrich) at a final concentration of 10 mM, followed by 30 minutes of incubation at 37 °C. From this reaction, 50 μg of total protein (~1/3 of the total) was removed and digested with TPCK treated trypsin (Sigma Aldrich) and Lys-C protease (Pierce) at an enzyme to protein ratio of 1:10 for trypsin and 1:20 for LysC. The digestion was left to proceed overnight at 37 °C. On the second day, samples were acidified with 10% TFA (Trifluoroacetic acid, ThermoFisher) to precipitate sodium deoxycholate

and stop the proteolysis. Samples were incubated for 15 min at room temperature and then centrifuged at $18,000 \times g$ for 10 minutes. The supernatant was then desalted using solid phase extraction HLB matrix (Oasis HLB μelution plate plate, 30 μm) with 0.1% FA (Formic acid) for the first wash and 5% methanol 0.1% TFA for the second wash. Samples were eluted with 75%ACN (Acetonitrile)/0.1% FA, dried using a centrifugal vacuum evaporator (LabConco) and stored at −20 °C.

**LC-MS Method.** Digested peptides were resuspended with 0.1% FA at a concentration of 400 ng μL-1 and 1 μg was injected directly on to an analytical Aurora C18 column (15 cm × 75 μm ID, 1.6 μm C18, Ionopticks, Australia) using an Easy-nLC 1200 UHPLC system (Thermo Fischer). The column temperature was maintained at 60 °C with an integrated column oven equipped with a spray adaptor and larger chamber to accommodate the fittings of the column (Sonation, Biberach, Germany, PROSO-V1). The LC gradient was generated using 0.1% formic acid in ddH2O, buffer A, and 80% acetonitrile 0.1% FA (SigmaAldrich), buffer B, linear gradient from 2–40% B over 60 minutes followed by an increase to 80% B in 2 minutes with a 4 min hold at 80% B before re-equilibration at 2% B prior to the next sample injection. The flow rate was 400 nL*min-1. The samples were analyzed on an Eclipse Tribrid Orbitrap mass spectrometer (Thermo Fisher) equipped with a FAIMS Pro source (Thermo Fisher). Data dependent LC-MS/MS was conducted at, the Spray voltage was set to 2–2.5 kV, FAIMS CV voltages of −50 and −70, RF funnel 40 V, heated capillary 305 °C. The instrument was operated in data-dependent mode with full profile MS spectra collected at 120k resolution (at 200 m/z), 375–1500 m/z, AGC target of 250% with maximum injection time set to Auto. The data-dependent acquisition used a monoisotopic peak determination filter set to peptides, intensity set to 5e3, charge states 2–6, dynamic exclusion of 60 seconds and a total cycle time of 0.6 seconds for each FAIMS CV voltage. The MS/MS fragmentation was collected with high energy collision induced dissociation (HCD) with the following settings; quadrupole isolation with 1.2 Da window, normalized collision energy set to 30%, detector set to ion trap with turbo scan rate, and defined mass range of 200–1400, AGC 250% with a maximum injection time of 25 ms. Information is stored as centroid data.

**Data analysis.** Proteomic data was analyzed using the commercially available software Proteome Discoverer version 2.4 (ThermoFisher). The LC-MS/MS files were recalibrated and MS/MS spectra were searched with Sequest using the following parameters; protein database homo sapiens (uniprot taxon ID9606 reviewed with 20,385 sequence entries), N-terminal modifications Met-loss, met-loss and acetylation, peptides fixed modifications carbamidomethylation of cys, dynamic modification included oxidation of Met, and pyroglutamate formation of N-terminal glu, Enzyme was set to trypsin with 2 miss cleavages, 10 ppm error on MS1 and 0.5 Da error on MS2. All spectra not matched were then re-processed using trypsin with semi tryptic cleavage and 2 missed cleavages. All data were passed through percolator with FDR set at 1%. Label free quantitation was done using the standard workflow provided with default settings, except the retention time filter was reduced to 3 minutes, due to the high reproducibility of the retention time using the current LC setup. MS1 features were extracted using minor feature detection and. ANOVA statistical analysis was done using pairwise protein-based abundance with the ratio of COVID positive/negative control samples. A Benjamini-Hochberg correction for FDR was applied and adjusted $p$-values < 0.01 and an abundance ratio of at least ± 1.3.

## Lipidomics and metabolomics
For all lipidomics and metabolomics experiments, patient samples were de-identified prior to receipt. The subsequent analysis of lipids was blinded. All patient samples were processed in a cold room at 4 °C.

## Untargeted lipidomics
**Quality control and internal standards.** Pooled quality control samples were prepared by aliquoting 5 μl of each patient plasma into a single vial. This sample was spiked with 10 μl internal standard (Splash Lipidomix, Avanti Polar, Birmingham, AL) and extracted in the same manner as patient samples. This analytical grade standard contains odd-chain, deuterated lipids in lipid classes and ratios present in human plasma. Pooled QC samples were run after every 10 patient samples as well as at the beginning and end of the analytical run. Quality control samples were used to ensure the stability of the instrument during the course of analysis. The coefficient of variation for each identified lipid within the pooled QC was then calculated using a cutoff of <40%. The internal standard was used to optimize instrumental parameters, such as electrospray voltage, collision energy, and others; these parameters were held consistent over the course of analysis. Lipids in the internal standard were also used to monitor injection consistency from sample to sample.

**Extraction.** Lipids were extracted from plasma using a high throughput, monophasic, methyl t-butyl ether (MtBE) based method. Using an automated pipetting and sample preparation system (Biotage Extrahera, Uppsala, Sweden), 50 μL of patient plasma was loaded into preconditioned wells containing 10 μL methanol and 10 μL of internal standard, Splash Lipidomix (Avanti Polar, Birmingham, AL). To each well, 200 μL methanol containing 50 μg/mL BHT was then added and the sample was mixed by 3 up and down passes of the automated sample handling pipette. The samples were then centrifuged at $1750 \times g$ for 5 minutes to pellet precipitated protein. The supernatant was recovered and transferred to a separate deep well 96-well plate for extraction. To extract lipids from the supernatant, 250 μL MtBE: methanol (3:1 v/v) was added to all wells and mixed with 3 up and down passes of the automated sample handling pipette. The sample plate was then centrifuged at $1000 \times g$ for 3 minutes and the supernatant was filtered through a 0.25 μm polytetrafluoroethylene (PFTE) filter plate (Biotage, ISOLUTE® FILTER+, Uppsala, Sweden) The recovered extract was then dried under nitrogen gas and subsequently reconstituted to 200 μl in acetonitrile: isopropanol (1:1 v/v) methanol for LC/MS analysis.

**Chromatography.** Ten microliters of extracted lipids were resolved on a Vanquish UHPLC (Thermo Scientific, Waltham, MA) using a Thermo Scientific Accucore C18 (4.6 × 100 mm, 2.6 μm) column on a 15 minute linear gradient, whereby Solvent A was 60:40 acetonitrile: water and Solvent B 90:10 isopropanol: acetonitrile. Both solvents in the mobile phase contained 0.1% formic acid and 10 mM ammonium formate. The column temperature was set at 50 °C and a flow rate of 0.4 mL/min was constant throughout analysis. Chromatography parameters are shown in Table S6.

**Mass spectrometry.** Eluted lipids were analyzed by a Thermo IDX Fusion mass spectrometer operated in both the positive and negative ionization modes successively (Thermo Scientific, Waltham, MA). To complete the untargeted lipidomics analysis, a data-dependent acquisition method was used. For this, a high-resolution MS scan was conducted on each sample at 120,000 FWHM resolution and ions above the instrumental noise threshold were systematically fragmented for structural elucidation. All MS/MS spectra were conducted using 30,000 FWHM resolution. Instrumental parameters used during analysis were optimized using the pooled quality control sample and the analytical grade internal standard. Parameters were held constant over the course of the analysis. Recorded instrumental parameters are provided in Table S7.

**Lipid identification.** Raw mass spectral data was uploaded into Lipid-Search software version 4.2 (Thermo, San Jose, CA) for lipid

identification (Table S8). This software scans user data against an extensive database to identify lipids. Peaks were detected and quantified using the QEX product ion search parameters, where 5.0 ppm was used as both parent and product mass tolerances. Other LipidSearch parameters, such as the selected database, retention time tolerance, precursor and product mass tolerances, intensity threshold, m-score threshold, ID quality filters, lipid class, and adducts, are provided in the Supplementary Information. For alignment, the pooled QC was designated as the control and its' fragment data used for identification of lipid species. The identified features were then aligned with the full scan data from each respective patient sample using a 0.1 minute retention time tolerance. XCMS software was used to process raw data files and identify features in the samples. Lipids were annotated with LipidSearch 4.2 software (Thermo). Only MS2 level confirmed lipid species grade A, B, and, C were used and lipids with grade D or lower were removed. Grade "A" calls are lipids of which fatty acid chains and class were identified completely, grade "B" calls are lipids of which class and some fatty acid chains were identified, grade "C" calls are lipids of which class or fatty acid was identified, while low confidence "D" identifications are only matched according to mass. Search parameters used for the LipidSearch software are provided in the Supplementary Information (Table S9). All preliminary identifications made by the software were manually reviewed to ensure appropriate identification and quantitation.

### Polyunsaturated fatty acids

**Extraction.** Polyunsaturated Fatty Acids (PUFAs) were extracted according to an adapted Bligh and Dyer method. For this, 1.2 mL of methanol and 0.6 ml chloroform was added to 100 μL patient plasma of sample. Samples were vortexed for 30 minutes and subsequently centrifuged at 1750 × g for 10 minutes at 4 °C. The supernatant was collected and 1 mL of 0.1 M Sodium Chloride and 1 mL of chloroform was added to each sample. Samples were then vortexed and centrifuged as above. The organic phase was retained and dried under a gentle stream of nitrogen gas. Dried samples were reconstituted in 100 μL 1:1 chloroform/methanol and 10 μL of Splash Mix (Avanti Polar, Alabaster, AL) was added. This sample was transferred to a HPLC vial for LC/MS analysis.

**Chromatography.** Extracted lipids were resolved using an Exion AC UPLC (Sciex, Framingham, MA), whereby 10 μL of the extract was deposited on an Accucore C18 column (100 × 4.6 mm; Thermo Waltham, MA) and resolved on an 18-minute linear gradient. For the mobile phase, 60:40 acetonitrile: water was used as Solvent A and 90:10 isopropanol: acetonitrile as Solvent B, both containing 10 mM ammonium formate and 0.1% formic acid. The column temperature was set to 50 °C and a 0.5 mL/min flow rate was held consistent over the course of the run. Chromatography parameters are shown in Table S10.

**Mass spectrometry.** Eluted lipids were analyzed by a Qtrap 5500 (Sciex, Framingham, MA). For this, precursor ion scanning was conducted in the negative ion mode for m/z 277, m/z 279, m/z 301, m/z 303, and m/z 327, corresponding to the mass-to-charge ratio of linolenic (18:3), linoleic (18:2), eicosapentaenoic (20:5), arachidonic (20:4), and docosahexaenoic (22:6) fatty acyl anions. Using an independent data acquisition (IDA) method, all resulting ions from each precursor ion scan within the mass range of m/z 200-m/z 1000 with signal intensities greater than 500 cps were fragmented for structural elucidation in a top 4 manner. Instrumental parameters were optimized using the internal standard and held consistent during analysis. Mass spectrometry parameters are presented in Table S11.

**Identification and quantification.** Lipids were identified using LipidView (Table S9) identification software (Sciex, Waltham, MA). This

software scans fragment data against an extensive database. After upload of raw data, lipids are identified by matching MS/MS data. Preliminary identifications were made by default "confirmed and common" search parameters and verified by manual review of product ion data. Quantification of identified lipids was achieved by single point calibration, whereby the area of the analyte was divided by the area of the internal standard and multiplied by the concentration of the internal standard, 15:0-18:1D7 PC. Standard lipidomics nomenclature was maintained throughout (x:y), where x is the number of acyl carbons and y are the number of double bonds. Ether-linked lipids are denoted as e; those with no qualifier are acyl linkages.

### Oxylipins and endocannabinoids (OXY)

**Standards.** Oxylipins and endocannabinoids are quantified by an external calibration curve using analytical grade standards, all purchased from Cayman Chemical (Ann Arbor, MI). These are PGE2 Ethanolamide (PGE2-EA), Oleoyl Ethanolamide (OEA), Palmitoyl Ethanolamide (PEA), ARA-Ethanolamide (AEA), Docosahexaenoyl Ethanolamide (DHEA), Linoleoyl Ethanolamide (LEA), Stearoyl Ethanolamide (ceramid), oxy-Arachidonoyl Ethanolamide (oxy-AEA), 2-Arachidonoyl Glycerol (2AG), Docosatetraenoyl Ethanolamide (DEA), alpha-linolenoyl ethanolamide(ALEA), 9Z-octadecenamide (oleamide), dihomo-gamma-linolenoyl ethanolamide, docosanoyl ethanolamide, (±)9,10-dihydroxy-12Z-octadecenoic acid (9,10 DIHOME), prostaglandin E2 (PGE E2), 20-hydroxy-5Z,8Z,11Z,14Z-eicosatetraenoic acid (20- HETE), (±)-9-hydroxy-5Z,7E,11Z,14Z-eicosatetraenoic acid (9-HETE), (±)14,15-dihydroxy-5Z,8Z,11Z-eicosatrienoic acid (14,15 DHET), (±)5-hydroxy-6E,8Z,11Z,14Z-eicosatetraenoic acid (5 HETE), 12R-hydroxy-5Z,8Z,10E,14Z-eicosatetraenoic acid (12 R-HETE), (±)11,12-dihydroxy-5Z,8Z,14Z-eicosatrienoic acid (11,12-DHET), (±)8,9-dihydroxy-5Z,11Z,14Z-eicosatrienoic acid (8,9-DHET), (±)5,6-epoxy-8Z,11Z,14Z-eicosatrienoic acid (5,6 EET), (±)5,6-dihydroxy-8Z,11Z,14Z-eicosatrienoic acid (5,6-DHET), Thromboxane (TXB2), (±)12(13)epoxy-9Z-octadecenoic acid (12(13)-EPOME), (±)-13-hydroxy-9Z,11E-octadecadienoic acid (13 HODE), prostaglandin F2a (PGF2A), (±)14(15)-epoxy-5Z,8Z,11Z-eicosatrienoic acid (14(15)-EET), (±)15-hydroxy-5Z,8Z,11Z,13E-eicosatetraenoic acid (15-HETE), Leukotriene B4 (LTB4), (±)8,9-epoxy-5Z,11Z,14Z-eicosatrienoic acid (8(9)-EET), (±)11,(12)-epoxy-5Z,8Z,14Z-eicosatrienoic acid (11(12)-EET). Calibration curves were prepared of these standards over the linear range of 0.1-10 nM and the equation of the slope used for quantification of the corresponding lipid in the patient's plasma.

**Extraction.** Oxilipins were isolated from plasma samples using an automated C18 solid phase extraction manifold, the Biotage Extrahera (Biotage, Uppsala, Sweden). Samples were prepared by pipetting 100 μL of patient plasma to a 96 well plate. To the sample, 300 μL of 20:80 methanol: water, 55 μL 1% butylated hydroxytoluene (BHT), and 80 μL of glacial acetic acid was added to achieve pH 3.0. Samples were centrifuged at 1750 × g for 10 minutes at 4 °C to pellet any precipitated solids. The supernatant was transferred to a C18 SPE plate (Isolute C18, Biotage, Uppsala, Sweden) that was preconditioned with 1 mL ethyl acetate and 1 mL of 95:5 water: methanol. After depositing the sample on the SPE plate, the sample was rinsed with 800 μL water, followed by 800 μL hexane. The oxylipins were then eluted off the SPE column with 400 μL methyl formate. The recovered oxylipin fraction was then dried under nitrogen gas and subsequently reconstituted in 100 μl methanol to be analyzed by LC/MS.

**Chromatography.** Extracted lipids were resolved using an Exion LC/QTrap 5500 LC/MS system (Sciex, Framingham, MA). To quantify the oxylipins in extract, 10 μL sample was injected onto an Accucore C18 column (100 × 4.6, Thermo, Waltham, MA) and resolved on a 16-minute gradient using water as Solvent A and acetonitrile as Solvent B, both containing 0.1% formic acid. The column was heated to 50 °C in a

temperature-controlled column chamber and 0.5 ml/min flow rate was used for analysis. The gradient used for analysis is shown in Table S12.

To detect endocannabinoids in the extract, a shortened chromatography gradient was programmed utilizing the same analytical column and mobile phase referenced above. For endocannabinoids, 1 μL of extract was injected into the column for analysis and resolved on a 13-minute linear gradient. Gradient parameters are provided in Table S13.

**Mass spectrometry.** Eluted oxylipins and endocannabinoids were analyzed by a QTrap5500 (Sciex, Framingham, MA) mass spectrometer. Instrumental parameters were optimized using external analytical grade standards and were held consistent over the course of analysis. Instrumental parameters are reported below. Oxylipins were analyzed in the negative ion mode and endocannabinoids were analyzed in the positive ion mode in successive analytical runs. Both classes of lipids were targeted using a multiple reaction monitored (MRM) based method. For this, the mass of the target lipid and a characteristic fragment were targeted for detection. Tables used for analysis are reported as Table S14 and Table S15.

**Quantification.** To quantify oxylipins and endocannabinoids, the area under the curve of each identified lipid is calibrated against an external calibration curve. To create the curve, analytical grade standards in the linear range of 0.1 nM–10 nM are created for each standard. The lower limits of detection and upper limits of detection are determined, plotting concentration versus area under the curve. The slope of the linear regression equation is used for calibration of the corresponding analyte.

## Untargeted metabolomics

**Extraction and quality control.** Metabolites were extracted from patient plasma using a 1:1 mixture of acetonitrile:methanol. Two-hundred μL 1:1 acetonitrile:methanol was added to 50 μl plasma. This solution was vortexed for 3 seconds and incubated on ice for 30 minutes. The sample was then centrifuged at 20,000 × g for 10 minutes at 4 °C to pellet precipitated protein. The supernatant was then transferred to an amber autosampler vial for LC-MS/MS analysis. A pooled quality control sample was created by combining 5 μl of each sample extract into a separate vial. This sample was run in triplicate and the beginning and end of analysis, as well as intermittently over the course of analysis. This was done as an analytical control and to assess instrument suitability. Additionally, the NIST Standard Reference Material 1950 (metabolites in human plasma) was injected as a global reference material to assess performance between studies. Both the global QC and the Standard Reference Material 1950 were also used to generate MS/MS scans using data-dependent acquisition as described below.

**Chromatography.** Untargeted metabolomics was performed using an ID-X Fusion (Thermo, San Jose, CA) tribrid mass spectrometer coupled to a Vanquish UHPLC (Thermo, San Jose, CA). Metabolites in the sample extracts were injected and resolved using a SeQuant ZIC-HILIC (3.5 μm, 100 A 150 × 2.1 mm) column with a 23-minute linear gradient method (detailed below). For chromatography, water was used as Solvent A and acetonitrile as Solvent B; both contained 0.1% formic acid. For analysis, 1 μl extract was injected by the autosampler and introduced to the LC column. The column was maintained at 50 °C. HESI-II source parameters are listed below. A full scan MS1 spectrum was obtained for each sample, including the pooled QC and procedural blanks, at 120,000 FWHM resolution within a scan range of $m/z$ 67–1000. The mass spectrometer was operated in both positive and negative ionization modes (as separate injections). Additional MS/MS scans were acquired in separate runs with data-dependent acquisition to support the identification of metabolites. The top 20 ion intensities

per MS1 scan were selected for fragmentation, generating substructure information. This process was done iteratively across repeated injections of the QC materials, excluding prior selections to increase coverage. Tables related to this analysis are included as Tables S16 and S17.

**Metabolite identification.** After collection, data were processed using Thermo Compound Discoverer software version 3.2 (San Jose, CA) (Table S18). Metabolites from the IROA Mass Spectrometry Metabolite Library of Standards (MilliporeSigma) were analyzed to generate a compound library of accurate mass and retention time data for the method described above. Metabolites were subsequently identified by accurate mass and retention time (denoted as 'AMRT'), following data alignment and peak integration by the Compound Discoverer software. Additional annotations were made by comparing MS2 results to the mzCloud library (mzcloud.org) of reference MS2 spectra (denoted as 'MS2L'). Compounds detected were compared to a compound library of exact mass and retention time of 413 validated reference standards, i.e., Schymanski level 1[89], and/or were annotated according to sufficient substructure information as determined from MS/MS scans, i.e., Schymanski level 2. If AMRT and MS2L designations for the same compound produced conflicting results, the reference standard data took precedence. Signal intensities were normalized by batch drift correction using the pooled QC sample and the maximum RSD after normalization was set at 30% (filtering those compounds with greater RSD), while compounds with a ratio of the max sample area to max blank area less than 5 were also filtered.

## Targeted metabolomics

To validate the metabolic pathway enrichments identified in untargeted LC-MS data, a targeted MS/MS approach combining LC-MS/MS and flow injection analysis (FIA-MS/MS) using reference compounds (Biocrates Life Sciences) was employed. This allowed for metabolite identification and quantification at level 1 (matching accurate mass, retention time, and MS/MS relative to authentic standards).

## Microfluidics assays

All microfluidics devices were generated in house using a standard workflow employed by our group beginning with soft photolithography and a silicon wafer etched with custom device design[90]. For each of the three microfluidics device designs used in the described experiments, three dimensional channels were constructed by first casting the wafer with polydimethylsiloxane (PDMS, Dow Corning) and then cured overnight at 60 °C. Casts were then separated from the mold and cylindrical biopsy punches were used to create inlet and outlet sites. For the microfluidics assays involving endothelial cell culture, each device was bonded to a second thin sheet of PDMS and for the aggregation and deformability assays, devices were bonded to cover glass slips (VWR) using a plasma bonder (PlasmaTech) and kept at 60 °C for at least 24 h before use.

**Shear and conduit size dependent RBC aggregation.** RBC aggregation under flow was assessed using a device consisting of two 500 μm bypass channels surrounding the central straight conduits. These devices contain duplicate copies of four channel widths: 20 μm, 30 μm, 45 μm and 70 μm, each with a height of 10 μm. Resistance is equal among all channels to permit equal distribution of flow. Only the 70 μm channel was used for aggregation analysis in these experiments. Each device was first coated with 1% bovine serum albumin (BSA, Sigma) in phosphate buffered saline (PBS) for one hour at room temperature to limit potential interaction between the blood sample and the device

material. For all tested conditions and patient samples, solutions were perfused into devices using a flow controlled syringe pump (Harvard Apparatus) to a target shear rate of $100\,s^{-1}$ based on a solution to laminar flow in a microfluidic rectangular cross sectional tube. RBC aggregation was quantified as the number of differential cell velocity clusters (DCVCs) using video microscopy coupled with a custom post-processing workflow. A Grasshopper 3 high speed camera (FLIR) was used to obtain 10 seconds of video frame data at a frame rate of 160 frames per second, with a FOV of $1200 \times 1920$ pixels and image resolution of 2.57 pixels/μm. Each of these video of 1600 frames was discretized into 40 sets of 40 consecutive frames for the tracking function, spanning 0.25 seconds. This time frame was chosen because the average particle velocity would be approximately 200 μm/s at this flow rate and 0.25 seconds would yield average displacement of 50 μm which is slightly greater than the expected length of an endothelial cell in the direction of flow. For the first frame of each set, cell features were identified using an algorithm to identify corner points or edges[91] which detects areas of an image with sharp gradients in image intensity. The Kanade-Lucas-Tomasi feature tracking algorithm was then used in Matlab (Mathworks) to follow the pathline for each of these cell features across the next 39 frames in the set using forward and backward validity checks set to maximum fidelity[92]. This would ensure that any noise within the image would not contribute to the profile of tracked points. This method for tracking velocity in microfluidics devices using red cell features has been previously published and was adapted for purposes in this study[39].

Figure S3 details the described workflow. A velocity field for each frame was generated from the tracking algorithm (SFig 3B) and instantaneous velocity for each tracked point was averaged over all frames within the set. The cumulative average velocity field was then discretized into $5\,μm \times 5\,μm$ bins to create a standardized velocity matrix (SFig 3C). The difference in velocity between neighboring bins was calculated and normalized to the average velocity of that frame set (SFig 3D). A threshold was applied to this differential velocity field map to identify all adjacent bins that had velocities which were different by less that 5% of the mean velocity to create a logical image of similar (map value = 1) or dissimilar (map value = 0). A connectivity map was then generated in which all contiguous areas meeting the defined threshold were identified (SFig 3E). All areas that were at least 115 pixels$^2$ large in this differential velocity map, corresponding to an area roughly the size of 15 RBCs, were considered a DCVC and the total number of DCVCs over the length of the video was quantified. The size of each DCVC was then calculated in image coordinates and the position of these clusters in the differential velocity map was mapped to the original velocity field to calculate the DCVC's velocity. This definition was chosen to identify groups of tracked cells that moved together as a cluster under flow-mediated either by interactions between RBC membranes or by differential changes in the biophysical properties of the blood from changes in the plasma viscosity or circulating proteins. The results were compared between patients with COVID and volunteers for both channel sizes and each target shear rate using an unpaired t-test between the two populations and repeated measures ANOVA within each population for the two channel sizes.

For the variable fibrinogen dose aggregation experiments, purified fibrinogen (Enzyme Research Laboratory) was diluted in PBS to concentrations of 900 mg/dL, 600 mg/dL, and 300 mg/dL. A negative control without fibrinogen was also tested. These dilutions were combined with RBCs isolated from healthy control volunteer. Blood was collected into a blood collection tube containing citrated anticoagulant. The platelet-rich plasma was removed from the red cells by the first centrifugation step and then the RBCs were washed of all residual plasma twice by the addition of PBS at twice the volume and repeated centrifugation steps. For the patient and healthy plasma aggregation experiments, plasma was collected, processed and stored as described above.

**RBC deformability.** To evaluate changes in RBC membrane properties, a previously described RBC deformability assay (Fig. 3E, top left) was used[93]. This assay allows visualization and measurement of the velocity of individual RBCs moving through a channel measuring $5\,μm \times 6\,μm$, the scale of the capillary, as a surrogate for membrane deformability. RBCs were isolated from each patient or volunteer on the day of collection as described above. The RBCs were then resuspended in PBS at 0.5% by volume and perfused at a steady flow rate of 1 μL/min using a flow-controlled syringe pump. Two minutes of video microscopy at a frame rate of 40 frames per second with image resolution of 2.57 pixels/μm was obtained to capture deformability data for at least 250 cells for each patient or volunteer. To determine a deformability index individual RBC velocity was calculated using at least three video frames of the cell's movement across the channel and then normalized to a mean reference value calculated by measuring the velocity of the cells through a larger channel in the device representing unobstructed flow (Fig. 3D, top left). The deformability indices for each patient or volunteer were pooled within the respective populations and the results were compared between patients with COVID + , non-COVID sepsis and healthy controls. The average number of cells evaluated across all subjects from the three cohorts was $735 \pm 505$.

**Glycocalyx degradation in endothelialized microfluidics devices.** To investigate the interaction between RBC aggregation and endothelial activation, a microfluidics platform was constructed with serial branches ranging in width from 120 μm to 30 μm with a constant height of 30 um (Fig. 2G) mimicking the approximate size and geometry of the pericapillary vasculature. Human umbilical vein endothelial cells (HUVECs, passage 4−7, Lonza, CC-2519) were cultured in this device using a well-established protocol[94,95]. Devices were first with fibronectin 50 μg/mL (Sigma) for 1 h at 37 °C and then washed with endothelial cell growth basal media (EBM) supplemented with EGM-2 Endothelial SingleQuots (Lonza). HUVECs grown to confluence in T25 flasks were washed twice with PBS, then detached from the flask by the addition of trypsin-EDTA and neutralized with EGM-2 media. Cells were pelleted by centrifugation, excess media was aspirated and cells were resuspended in 65−85 μL of EGM-2 with 8% dextran (500 kDa, Sigma) weight by volume to increase fluid viscosity and promote settling of cells. This cell resuspending was injected into the device inlet using negative pressure applied at the outlet of the device. Cells were allowed to adhere, and the inlets and outlets were cleaned of excessive cells. Cells were then cultured under steady flow using a flow-controlled syringe pump to target shear stress of 5 dyn/cm$^2$ and reached confluence by 72 h for use in the experiments.

For the variable fibrinogen dose experiments, fibrinogen was diluted to its final concentration in EGM-2 with the volumes of the lower fibrinogen levels normalized using PBS so that all conditions had identical volumes of cell culture media.

For the experiments comparing the influence of plasma and serum from COVID patients, plasma was collected in citrated blood collection tubes and serum was created by the addition of calcium to a final concentration of 10 mM. Clot formation was allowed to occur and separated by high rotational centrifugation. The resultant supernatant was then used as a serum.

For the patient and volunteer sample experiments, plasma was isolated from blood in citrated collection tubes from eleven patients

with COVID, five patients with sepsis from infectious causes other than SARS-Cov-2, and five healthy volunteers.

In each of these experiments, RBCs obtained from a healthy volunteer and isolated as above were combined with the fibrinogen dilutions, plasma, or serum to 25% by volume. The experiments involving plasma required samples to be re-calcified to 4 mM Ca to maintain endothelial cell homeostasis. In the case of serum and the fibrinogen dilution the calcium concentration was adequate either through the addition of cell culture media or the additional calcium supplementation. Devices cultured with confluent endothelial cells were perfused with Alex Flour 647 wheat germ agglutinin (Invitrogen) 10 μg/mL in EGM-2 for 1 hour and then washed with EGM-2 for 30 minutes. The devices were then imaged on a digital microscope (Keyence, BZ-X series) using high sensitivity resolution and an exposure time 1/15 seconds using a Cy-5 filter (Keyence). Tile scans of both the fluorescent and bright field images were obtained. The fluorescent signal was quantified in each channel size after an adaptive background subtraction was applied. The combined fibrinogen dilution/plasma/serum and RBCs were then perfused into the device at a steady rate using a flow-controlled syringe pump for 30 minutes. The imaging process was repeated using identical acquisition parameters. The pre and post-perfusion images were co-registered and the fractional residual glycocalyx was calculated as ratio of fluorescent signal in each channel segment and stratified by channel size. The residual glycocalyx ratio was compared across all channel sizes between the experiments using a paired repeated measures ANOVA test.

### Plasma enzyme-linked immunoassays
Plasma samples were processed and handled according to the manufacturer's instructions for each of the assays: syndecan-1 (abcam, ab46506), vWF (abcam, ab223864). All samples and standards were run in duplicate for each marker of endothelial dysfunction.

### Statistics and reproducibility
In all experiments, analyses were performed by applying appropriate statistical tests as detailed in the Methods subsections above and in Figure Legends. No statistical method was used to predetermine the sample size. All samples were de-identified prior to receipt for proteomics, lipidomics, and metabolomics experiments, and subsequent experimentation and analysis were blinded. Correlation assessments were performed using Pearson correlation coefficient and $p$-value < 0.05. For proteomics, ANOVA statistical analysis was done using pairwise protein-based abundance with the ratio of COVID positive/negative control samples. A Benjamini-Hochberg correction for FDR was applied and adjusted $p$-values < 0.01 and an abundance ratio of at least ±1.3. For metabolite and lipidomics analysis, data from each assay (Biocrates Quant500, Oxylipins, PUFA, Untargeted high-resolution lipidomics and Untargeted high-resolution metabolomics) were independently analyzed using the R package, xmsPANDA (https://github.com/kuppal2/xmsPANDA). Missing values were imputed using the k-nearest neighbors/half minimum feature method and data were normalized by z-score. Significant features ($p$-value < 0.05) were identified using Limma (linear models for microarray data). For adults, significant features were further filtered by covariate analysis including gender ($p$ value_gender < 0.05) and age ($p$ value_age < 0.05). KNN nearest neighbor method was used to impute values identified in some patients but not others. Significant metabolomics features from each platform were merged and duplicates were cleaned. Pathway analysis was performed using MetaboAnalyst (version 5.0), also integrating proteomics data sets[96]. Z-Normalization was used to compare different methods. This method is a metabolite-based normalization method to adjust the metabolite variances identified via our suit of metabolomics/lipidomics analysis. Metabolites are comparably scaled to unit variance based on the assumption that all metabolites are equally important. Z-score is calculated for each metabolite as $(x-\mu)/\sigma$, where $x$ is the metabolite (unit: concentration or area), $\mu$ is the mean of the metabolite, and $\sigma$ is the standard deviation of the metabolite. For cytokine and metabolomics data integration, statically significant metabolomics data were pre-filtered to retain only features with non-zero values in >50% in all samples and >80% in each group. The R package xMWAS (https://github.com/kuppal2/xMWAS) was used to integrate cytokines and metabolomics data. Multivariate statistical method partial least squares regression (PLS) was used where correlation threshold is 0.4 and $p$-value threshold is 0.05. The integration network was evaluated by eigenvector centrality (importance) and nodes' centrality >0.4 were selected and their edges used to plot the heatmap.

### Reporting summary
Further information on research design is available in the Nature Portfolio Reporting Summary linked to this article.

## Data availability
Data reported in this study are publicly available. The mass spectrometry proteomics data have been deposited to the ProteomeXchange Consortium via the PRIDE partner repository with project accession PXD040437 for the pediatric cohort, and project accession PXD040438) for the adult cohort. MS/MS spectra in the study were searched as described above using uniprot taxon ID9606 (https://www.uniprot.org/taxonomy/9606). The lipidomics, metabolomics, and microfluidics data are provided in the Supplementary Information/Source Data files. All other source data are provided with this paper in the accompanying Source Data file. Source data are provided with this paper.

## Code availability
All code related to microfluidics raw data post-processing workflow can be found at https://github.com/lmiffrig/NatCommCOVID.

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

## Acknowledgements

The authors wish to acknowledge the members of Emory's COVID-19 Anticoagulation Working Group, led by Sara C. Auld, and the S1 Toxin Hypothesis Group. We appreciate the involvement of the Epidemiology of ARDS Biorepository, including Greg Martin, Annette Esper, and Phil Yang. The Emory Children's Center-Vaccine Research Clinic thanks Laila Hussaini, Ashley Tippett, Austin Lu, Caroline Ciric, Lisa Macoy, and Kathy Stephens for assistance with this study. We also thank the Emory and the Children's Healthcare of Atlanta participants and families, the clinical and research staff, and, indubitably, the many other people that made this study possible by their participation or support. Some figure panels included in the manuscript were created with BioRender.com. This study was made possible by financial support from a number of sources, including the following grants: F31 training fellowship from the National Institutes of Health (NIH) National Institute of Diabetes and Digestive and Kidney Diseases (NIDDK), F31DK126435, to support S.A.D.; Stimulating Access to Research in Residency from NIH, R38AI140299, to support E.I.; NIH/National Heart, Lung, and Blood Institute (NHLBI) K99 HL150626-01 awarded to C.L.M.; R35HL145000 supported E.I., Y.S., K.S.F. and W.A.L.; NIH/NHLBI HL150658 awarded to J.D.C.; a training grant supported by Emory's Biochemistry and Cell Developmental Biology program, T32GM135060-02S1, to S.O.K.; NIH/NIDDK Grant R01-DK115213 and Winship Synergy Award to E.A.O. We are grateful for financial support gifted through the Community Foundation to C.L.M., and the FastGrant Program from Emergent Ventures at the Mercatus Center at George Mason University and Genentech, Inc. to A.C. Finally, we thank the Emory Integrated Metabolomics and Lipidomics Core, Emory Integrated Biorepository Core, Emory University School of Medicine, Emory's Woodruff Health Sciences Center, Children's Healthcare of Atlanta, the Georgia Research Alliance, and the Donaldson Trust for their support. Finally, the content presented here is solely the responsibility of the authors and does not necessarily represent the views of the NIH or other agencies.

## Author contributions

C.L.M, G.S., D.A.K., B.R.R., W.A.L., and E.A.O. conceived, designed and oversaw the study. E.I. developed image processing workflows for microfluidics assays, designed and completed the red cell aggregation studies, deformability assays and glycocalyx degradation assays, performed the analysis for all experiments utilizing microfluidics, and generated all related figures. S.D. analyzed integrated multiomics data, assisted in untargeted metabolomics annotations using Compound Discoverer, and generated figures for metabolomics, proteomics, and lipidomics data analysis. K.S.F. manufactured microfluidics devices and assisted in glycocalyx degradation assays. Y.S. assisted with experimental and microfluidics device design as well as microscopy image collection. A.A.I. analyzed multiomics data. J.A. prepared and ran samples and assisted in the development of the untargeted metabolomics assay. X.L. collected and processed oxylipin and PUFAs data. K.M-S. processed and analyzed oxylipin and PUFA data. T.B. drafted all the compiled metabolomics methods. J.D.C. and S.K. designed the untargeted metabolomics assay and curated a compound library of reference standards for annotations. J.D.C. processed data using Compound Discoverer and assisted in data evaluation. B.R.R designed, performed, and analyzed cytokine and proteomic data sets. A.M.R. processed samples, performed proteomics experiments, and subsequent data processing. T.Z. integrated metabolomics, lipidomics, and cytokine information and performed statistical analysis. R.G., F.S., Mo.M., and Ma.M. performed autopsies and provided pathologic review. F.S. also performed immunohistochemical and electron microscopy analysis. J.D.R., H.P.V., I.A., S.R.S., and C.M.A. provided adult clinical samples, clinical data and analytical support. A.W., M.P., and S.E.B. provided input on experimental design and technical assistance. E.J.A., C.A.R., and A.C. provided pediatric specimens and clinical data as well as technical assistance. E.I., S.D., and C.L.M. wrote the manuscript, which was reviewed and commented on by all authors.

## Competing interests

E.J.A. has consulted for Pfizer, Sanofi Pasteur, Janssen, and Medscape, and his institution receives funds to conduct clinical research unrelated to this manuscript from MedImmune, Regeneron, PaxVax, Pfizer, GSK, Merck, Sanofi-Pasteur, Janssen, and Micron. He also serves on a safety monitoring board for Kentucky BioProcessing, Inc. and Sanofi Pasteur. His institution has also received funding from NIH to conduct clinical trials of Moderna and Janssen COVID-19 vaccines. C.A.R.'s institution has received funds to conduct clinical research unrelated to this manuscript from BioFire Inc, GSK, MedImmune, Micron, Janssen, Merck, Novavax, PaxVax, Regeneron, and Sanofi-Pasteur. She is co-inventor of patented RSV vaccine technology unrelated to this manuscript, which has been licensed to Meissa Vaccines, Inc. Her institution has received funds from NIH, Moderna, Pfizer, and Janssen to conduct clinical trials of COVID-19 vaccines. B.R.R. receives research support from Agilent, Bruker and eMSion for projects unrelated to this manuscript. The remaining authors declare no competing interests.

## Additional information

**Samuel Druzak**[1,14], **Elizabeth Iffrig**[2,3,14], **Blaine R. Roberts** ®[1,4], **Tiantian Zhang**[5], **Kirby S. Fibben**[3], **Yumiko Sakurai**[3,6], **Hans P. Verkerke**[7], **Christina A. Rostad**[6,8], **Ann Chahroudi** ®[6,8], **Frank Schneider**[7], **Andrew Kam Ho Wong**[7,9], **Anne M. Roberts**[1], **Joshua D. Chandler** ®[6,8], **Susan O. Kim**[6], **Mario Mosunjac**[7], **Marina Mosunjac**[7], **Rachel Geller**[7,10], **Igor Albizua**[7], **Sean R. Stowell**[11], **Connie M. Arthur**[11], **Evan J. Anderson** ®[2,6,8], **Anna A. Ivanova** ®[5], **Jun Ahn**[5], **Xueyun Liu**[5], **Kristal Maner-Smith**[5], **Thomas Bowen**[5], **Mirko Paiardini** ®[7,9], **Steve E. Bosinger** ®[2,7,9,12], **John D. Roback** ®[7], **Deanna A. Kulpa** ®[7,9,13], **Guido Silvestri** ®[7,9,12,13], **Wilbur A. Lam** ®[3,6,8] ✉, **Eric A. Ortlund** ®[1,5] ✉ & **Cheryl L. Maier** ®[7] ✉

---

[1]Department of Biochemistry, Emory University School of Medicine, Atlanta, GA, USA. [2]Department of Medicine, Emory University School of Medicine, Atlanta, GA, USA. [3]Wallace H Coulter Department of Biomedical Engineering, Georgia Institute of Technology and Emory University, Atlanta, GA, USA. [4]Department of Neurology, Emory University School of Medicine, Atlanta, GA, USA. [5]Emory Integrated Metabolomics and Lipidomics Core, Emory University School of Medicine, Atlanta, GA, USA. [6]Department of Pediatrics, Emory University School of Medicine, Atlanta, GA, USA. [7]Department of Pathology and Laboratory Medicine, Emory University School of Medicine, Atlanta, GA, USA. [8]Children's Healthcare of Atlanta, Atlanta, GA, USA. [9]Emory National Primate Research Center, Atlanta, GA, USA. [10]Georgia Bureau of Investigation, Decatur, GA, USA. [11]Department of Pathology, Brigham and Women's Hospital, Harvard Medical School, Boston, MA, USA. [12]Emory Vaccine Center, Atlanta, GA, USA. [13]Center for AIDS Research, Emory University, Atlanta, GA, USA. [14]These authors contributed equally: Samuel Druzak, Elizabeth Iffrig. ✉e-mail: wilbur.lam@emory.edu; eortlun@emory.edu; cheryl.maier@emory.edu

