## [Peer review file · Nature Communications]

REVIEWER COMMENTS

Reviewer #1 (Remarks to the Author):

Druzak et al describe the results from a multi-platform analysis of the drivers of pathogenesis in adult and pediatric COVID-19 patients. While previous studies have mostly focused on adult populations, to the best of this reviewer's knowledge this is the first report focusing on pediatric populations. While incidence of severe disease and hospitalization was extremely low with the original SARS-CoV-2 variant, increased disease severity and multi-inflammatory syndrome in children were noted especially upon infection with later variants. Here the authors provide perhaps the first mechanistic explanation of MISC, as result of red blood cell aggregation-induced damage to the endothelial glycocalyx. This finding is not only of interest to the overall medical community, but also describe a novel mechanism that could perhaps be of relevance to other infectious diseases or endotheliopathies. Other groups had originally described the damage to RBCs in the context of COVID-19, while this is the first report of the mechanistic relevance of this phenomenon in follow-up complications to COVID-19 in the pediatric patient population. The microfluidic device experiments are interesting and important. The studies on RBC incubation with COVID-19 plasma recapitulate studies from others (Rectkenwald et al – see specific comment below), but are not any less important. Similarly, the findings on unique cytokine signatures in adults and children with COVID-19 (with or without MISC) are of obvious biomedical relevance. Overall, this is a relevant and interesting study. Analyses are well-performed and data presentation is on par with the quality of the journal. Overall I am happy to recommend publication of the manuscript upon introduction of minor revisions as detailed below.

1) Duplicate references (e.g., 11 and 36 – latter is correct). Suggest replacing the first one with other plasma proteomics ones from the Espinosa lab (Proc Natl Acad Sci U S A . 2022 Mar 15;119(11):e2116730119. And/or Cell Rep. 2021 Aug 17;36(7):109527. And/or Elife. 2021 Mar 16;10:e65508. doi: 10.7554/eLife.65508.)

2) Line 164 - Previous reports have described changes in the lipidome and PUFAs as a function of patient age and BMI. Please, reference - Cells. 2021 Sep 2;10(9):2293. doi: 10.3390/cells10092293.

3) Studies by Recktenwald and colleagues showed that RBCs from COVID-19 patients, or from healthy subjects exposed to COVID-19 plasma, have morphological alterations, altered deformability and behavior in microfluidic devices. Please, comment on your data in light of these results (<https://doi.org/10.1101/2022.03.31.22273226>), especially given the results described in Figure 3.

Reviewer #2 (Remarks to the Author):

There is a growing body of evidence that various systemic diseases induced by severe COVID-19 are related to endothelial dysfunction and microvascular thrombosis. In this manuscript, the authors investigated its mechanism through a combined multiomics and microfluidic scheme. This is a very meaningful research project. On the one hand, the experimental results showed that fibrinogen induced RBC aggregation caused by severe COVID-19 was a newly discovered pathway leading to the damage of the endothelial glycocalyx. On the other hand, the results of its multiomics study showed that the changes in complement and coagulation pathways were important characteristics of severe COVID-19, which is consistent with the results of previous studies. This study also found different key mediators of pathogenesis between adult and pediatric populations. The findings of this study provide possible guidance for the treatment of severe COVID-19. However, this study involved a lot of data and analysis discussions, which made it not easy to understand. The following questions need to be well explained to help the reader better understand this work.

- 1、 Title: This study mainly investigates the samples of severe COVID-19. Is it necessary to show the keywords related to "severe acute" in the title?
- 2、 Figure 1: Multiomics Methods are used to analyze adult plasma samples. However, Figure 1 mainly shows the results of proteomics analysis, and the results of related lipidomic and metabolomic analyses are shown in Figure S1. If so, does Figure1A need to identify the parts of lipidomic and metabolomic analyses? An expression similar to Figure 4a is suggested.
- 3、 Line 155-157: 10 critically ill PCR-confirmed COVID-19-negative patients are selected as the reference group for the multiomics analysis, which is called Sepsis Cohort in Supplemental Table 1. However, their clinical diagnosis is not all sepsis. What is the selection principle of the reference group? Are there any common clinical features to all patients?
- 4、 The sample size used in this study is relatively small, but the conclusion obtained is mutually supportive with previous studies. How to evaluate its guiding significance for clinical application in such situations?
- 5、 Line 186-192: As described in the manuscript, many proteins are involved in remodeling of the endothelial glycocalyx. Why is the fibrinogen selected as the study object? I mean there should be more than one pathway that leads to red blood cell aggregation. Are there any other possible indications are given in the proteomics results? In addition, the authors draw the following conclusions: Fibrinogen mediated RBC aggregation induces endothelial glycocalyx degradation (Line 260-261). May non-fibrinogen-mediated RBC aggregation lead to endothelial glycocalyx degradation?
- 6、 Line 190-198: Since the regulated proteins in COVID+ patients are mainly located in the liver, does it mean that it is necessary to conduct in-depth research specifically on the liver to explore the drivers of systemic pathogenesis caused by severe COVID-19?
- 7、 Line 333-470: All of these experiments use the samples in Table s6, but not all of them seem to be selected in each experiment. Can the authors describe the principles of sample selection for each experiment in more detail? And why are the samples in Table S1 no longer used in these experiments? If possible, it is recommended that the same samples be used for all experiments for better comparability.

- 8、 Line 333-357: The results shown in Figure 3BCD don't seem to be able to tell the RBC aggregation difference between the plasma from COVID-19 patients and critically ill COVID-19-negative patients. Although Figure 3E may give a new finding that COVID-19 patients tend to have a wider range of DCVC velocities on an individual, the differences are not significant. Does it mean that although there are different pathways for COVID-19 sepsis, the main pathogenesis is similar to non-COVID-19 sepsis?
- 9、 Line 426-428: It seems that the authors do not give how the concentrations of these markers are determined.
- 10、 Line 457-460: Why are large, freshly collected citrated samples necessary in microfluidics experiments?
- 11、 Line 506-508: No significant increase in the abundance of fibrinogen chains is detected in acute COVID+ or MIS-C children. Are there other markers that cause RBC aggregation or endothelial glycocalyx degradation found?
- 12、 Line 547-550: With the development of mass spectrometry, there are many high-throughput methods for the identification of unsaturated fatty acids, such as the combination method of Waters ACQUITY UPC2 and PURSPEC Ω Analyzer. These methods may be helpful to the author's further research.
- 13、 Line 578-582: In Figure 6, the author focuses on the differences between children and adult patients. Are there some common markers for children and adult patients of severe COVID-19?
- 14、 Line 659-662: Why can the author come to such a conclusion that the destruction of the protective glycocalyx layer is not a result of enzymatic degradation?
- 15、 Line 1240-1380: In this work, a total of three microfluidic chips are used, and the chips used for RBC Deformability and Glycocalyx Degradation Research are very similar in structure, which leads me to mistakenly think that only two chips were used when reading the main text. In order to understand this part better, it is suggested that references should be given when the corresponding microfluidic chips are first mentioned in the main text (Figure 2C, Figure 2G and Figure 3F). And the structure diagram of each chip should be given in the Supplemental Literature.
- 16、 Line 1315: It should be Figure 3E instead of Figure 3D.

REVIEWER COMMENTS

Reviewer #1 (Remarks to the Author):

Druzak et al describe the results from a multi-platform analysis of the drivers of pathogenesis in adult and pediatric COVID-19 patients. While previous studies have mostly focused on adult populations, to the best of this reviewer's knowledge this is the first report focusing on pediatric populations. While incidence of severe disease and hospitalization was extremely low with the original SARS-CoV-2 variant, increased disease severity and multi-inflammatory syndrome in children were noted especially upon infection with later variants. Here the authors provide perhaps the first mechanistic explanation of MISC, as result of red blood cell aggregation-induced damage to the endothelial glycocalyx. This finding is not only of interest to the overall medical community, but also describe a novel mechanism that could perhaps be of relevance to other infectious diseases or endotheliopathies. Other groups had originally described the damage to RBCs in the context of COVID-19, while this is the first report of the mechanistic relevance of this phenomenon in follow-up complications to COVID-19 in the pediatric patient population. The microfluidic device experiments are interesting and important. The studies on RBC incubation with COVID-19 plasma recapitulate studies from others (Rectkenwald et al – see specific comment below), but are not any less important. Similarly, the findings on unique cytokine signatures in adults and children with COVID-19 (with our without MISC) are of obvious biomedical relevance. Overall, this is a relevant and interesting study. Analyses are well-performed and data presentation is on par with the quality of the journal. Overall I am happy to recommend publication of the manuscript upon introduction of minor revisions as detailed below.

We very much appreciate the reviewer's comments and interest in our study, as well as the opportunity to improve the manuscript by addressing the points raised by the reviewer below.

1) Duplicate references (e.g., 11 and 36 – latter is correct). Suggest replacing the first one with other plasma proteomics ones from the Espinosa lab (Proc Natl Acad Sci U S A . 2022 Mar 15;119(11):e2116730119. And/or Cell Rep. 2021 Aug 17;36(7):109527. And/or Elife. 2021 Mar 16;10:e65508. doi: 10.7554/eLife.65508.)

Thank you for catching the duplicate references and also for suggesting additional relevant references. We have corrected the duplication and replaced the first instance with the suggested Galbraith et al manuscript (Elife 2021), which highlights work from the body of literature evaluating novel pathways associated with the natural history and disease severity of COVID-19.

2) Line 164 - Previous reports have described changes in the lipidome and PUFAs as a function of patient age and BMI. Please, reference - Cells. 2021 Sep 2;10(9):2293. doi: 10.3390/cells10092293.

Thank you for this suggestion. We have included the reference and modified the text to reflect this point: "Our metabolomic and lipidomic analyses of adult and pediatric COVID-19 plasma similarly agree with previously reported findings, including changes in our adult cohort related to xanthine, acylcarnitines, PUFAs, and lactic acid. These metabolites and lipid by-products have been implicated previously in inflammatory processes, and, more recently, others have reported a positive correlation between certain PUFAs and increasing BMI or age in COVID-19 patients."

3) Studies by Recktenwald and colleagues showed that RBCs from COVID-19 patients, or from healthy subjects exposed to COVID-19 plasma, have morphological alterations, altered deformability and behavior in microfluidic devices. Please, comment on your data in light of these results (<https://doi.org/10.1101/2022.03.31.22273226>), especially given the results described in Figure 3.

Thank you for highlighting this recent study, which we have included in our revised manuscript. The study by Recktenwald and colleagues demonstrates changes in red blood cell (RBC) membrane properties following exposure to COVID-19 plasma as well as the normalization of such properties in RBCs from COVID-19 patients following exposure to healthy donor plasma. Overall, the Recktenwald et al study, which focuses on morphological changes of individual RBCs, is consistent with our findings of decreased RBC deformability in COVID-19. Notably, however, we observed relatively less RBC deformability with COVID-19 plasma than with non-COVID sepsis plasma, and future studies to investigate how the morphological changes described by Recktenwald et al in COVID compare to those in non-COVID sepsis or other critically ill states would be beneficial.

It is important to note that our study finds COVID-19 plasma induces RBC aggregation to such a pathologic degree that it withstands shear stresses and blood hemodynamics under physiologically-relevant flow conditions, ultimately leading to biomechanical degradation of the endothelial glycocalyx. This is quite distinct from the rouleaux formation of RBC reported by Recktenwald et al. Specifically, rouleaux occurs as a result of electrostatic forces that cause attraction (and stacking) of red blood cells under static conditions; however, under flow, shear forces overcome the electrostatic rouleaux forces. Here we report that COVID-induced RBC aggregation is strong enough to not only induce stable RBC clusters under physiologic flow conditions, but also to mechanically injure the endothelium. Overall, both studies underscore the important influence of plasma on the biophysical behavior of RBCs in COVID-19, with our study directly implicating fibrinogen and endothelial damage as key pathologic features. We have included the Recktenwald reference in our revised manuscript and added the following lines to our discussion to address this comment: “Disturbances in RBC biophysical properties – membrane shape and single cell flow behavior – have likewise been reported in recent literature and have been shown to occur following exposure to COVID-19 patient plasma. Notably, here we find that COVID-19 induces RBC aggregation to such a pathologic degree that RBC clusters are not only stable under physiologic flow conditions but also stable enough to cause microvascular injury, which is distinct from weaker electrostatic forces that may cause rouleaux formation of RBC under static conditions.”

Reviewer #2 (Remarks to the Author):

There is a growing body of evidence that various systemic diseases induced by severe COVID-19 are related to endothelial dysfunction and microvascular thrombosis. In this manuscript, the authors investigated its mechanism through a combined multiomics and microfluidic scheme. This is a very meaningful research project. On the one hand, the experimental results showed that fibrinogen induced RBC aggregation caused by severe COVID-19 was a newly discovered pathway leading to the damage of the endothelial glycocalyx. On the other hand, the results of its multiomics study showed that the changes in complement and coagulation pathways were important characteristics of severe COVID-19, which is consistent with the results of previous studies. This study also found different key mediators of pathogenesis between adult and pediatric populations. The findings of this study provide possible guidance for the treatment of severe COVID-19. However, this study involved a lot of data and analysis discussions, which made it not easy to understand. The following questions need to be well explained to help the reader better understand this work.

We would like to thank the reviewer for highlighting the significance of our study and for the very helpful feedback to improve the manuscript's clarity and provide better understanding for readers.

1、 Title: This study mainly investigates the samples of severe COVID-19. Is it necessary to show the keywords related to "severe acute" in the title?

Thank you for this suggestion. We have modified the title to include both suggested keywords.

2、 Figure 1: Multiomics Methods are used to analyze adult plasma samples. However, Figure 1 mainly shows the results of proteomics analysis, and the results of related lipidomic and metabolomic analyses are shown in Figure S1. If so, does Figure1A need to identify the parts of lipidomic and metabolomic analyses? An expression similar to Figure 4a is suggested.

We appreciate this suggestion as Figure 1B-H reflects all proteomic data. We have modified Figure 1A to remove mention of the lipidomic and metabolomic studies that are included in Figure S1, which harmonizes Figure 1 with Figure 4 as suggested.

3、 Line 155-157: 10 critically ill PCR-confirmed COVID-19-negative patients are selected as the reference group for the multiomics analysis, which is called Sepsis Cohort in Supplemental Table 1. However, their clinical diagnosis is not all sepsis. What is the selection principle of the reference group? Are there any common clinical features to all patients?

We appreciate the reviewer for catching this mistake, as the adult multiomics control group was mislabeled in the table (Supplemental Table 1). The label has been corrected to “Non-Covid Cohort” in the revision. The text describing this cohort within the manuscript is correct: “...Plasma was obtained from critically ill PCR-confirmed COVID-19-positive patients (COVID+, n=15) and from critically ill PCR-confirmed COVID-19-negative patients (COVID-, n=10) for multiomic investigation (Figure 1A). Notably, all patient samples in this adult cohort were obtained on a single day in April 2020, before the use of now standard therapies or the availability of vaccines. Patient demographics and relevant clinical characteristics are provided in the Supplemental materials (Table S1).” Thus, the adult patients used in the multiomics studies were all critically ill and being tested for SARS-CoV-2 given clinical suspicion for COVID (the common feature) on a single day at our institution. Those with a negative PCR result were assigned to the non-COVID group while those with a positive PCR result were assigned to the COVID+

group. Additional clinical features (underlying diagnoses, co-infections and co-morbidities) are provided in Supplemental Table 1.

4、 The sample size used in this study is relatively small, but the conclusion obtained is mutually supportive with previous studies. How to evaluate its guiding significance for clinical application in such situations?

This is an important point which must be considered any time relatively small cohorts are used. As the reviewer notes, our hypothesis-generating multiomic approach allowed us to identify specific changes and differential regulation supported by previous studies, suggesting that, despite a limited initial cohort, our methodology and results are robust in terms of biological and statistical significance. We fully acknowledge that this cohort was small, yet our goal was not to provide a comprehensive description of all aspects of COVID-19 but rather to identify pathways potentially involved in disease pathogenesis that could be further and more directly investigated through detailed mechanistic studies. Specifically, results from the multiomic studies guided our focus on unexpected changes in pathways related to endothelial cell biology. To this end, we performed a robust mechanistic investigation of altered blood rheology and red blood cell aggregation using microfluidic assays in a well-balanced and well-powered clinical cohort. We have modified the manuscript text to highlight this point more fully in the study limitations paragraph within the discussion section of the manuscript revision: “Future studies using larger patient cohorts, including pediatric patients with MIS-C or acute pediatric patients with significant hyperfibrinogenemia, are necessary to establish the role of altered blood rheology as a contributor to disease morbidity and to guide any clinical application, including interventions aimed at decreasing fibrinogen and/or normalizing blood rheology.”

5、 Line 186-192: As described in the manuscript, many proteins are involved in remodeling of the endothelial glycocalyx. Why is the fibrinogen selected as the study object? I mean there should be more than one pathway that leads to red blood cell aggregation. Are there any other possible indications are given in the proteomics results? In addition, the authors draw the following conclusions: Fibrinogen mediated RBC aggregation induces endothelial glycocalyx degradation (Line 260-261). May non-fibrinogen-mediated RBC aggregation lead to endothelial glycocalyx degradation?

We agree with this reviewer that there may be additional factors influencing red blood cell aggregation and the widespread endothelial damage seen in patients with COVID-19. Fibrinogen emerged as our primary target molecule for several reasons. First, fibrinogen is the primary mediator of red cell aggregation and the underlying reason patients with inflammation have an increased erythrocyte sedimentation rate (ESR), as the ESR is a marker of RBC-fibrinogen interactions. This is supported by the very strong positive correlation we found between ESR and fibrinogen in COVID patients (Figure 2A). In addition, levels of fibrinogen in COVID patients have been observed at unprecedentedly high levels, sometimes more than three times the upper limit of normal, which has been widely reported in the literature. Fibrinogen is one of the top three most abundant plasma proteins, alongside albumin and globulins. Interestingly, albumin has an inverse effect on red blood cell aggregation, with increasing levels causing less RBC aggregation, and does not appear to be significantly altered in COVID-19. Globulins are appreciated to have an effect on RBC interactions, most notably the rouleaux formation characterized by stacking of RBCs under static conditions, as in peripheral blood smears from patients

with plasma cell myeloma or hypergammaglobulinemia (please also see response to Reviewer 1 comment 3). However, we have not observed significant rouleaux formation in clinical blood smears from our COVID patients. Moreover, in unpublished clinical studies we have quantified the total amount of antibody in our COVID-19 patients, including anti-SARS-CoV-2 IgG and IgM as well as autoantibodies, finding that levels are within the normal range even during peak illness. Finally, our proteomics pathway analysis affirmed disturbances of both fibrinogen regulation and fluid shear stress pathways, which focused our efforts on the microfluidics experiments at the intersection of these two pathways. We have modified the manuscript by adding the following: “While we highlight the influence of fibrinogen on RBC behavior in COVID-19 and its effect on the endothelial glycocalyx, the contribution of additional pathways, including other circulating proteins promoting RBC aggregation or changes to RBC membranes at either the phospholipid or membrane protein level, remain to be explored.”

6、 Line 190-198: Since the regulated proteins in COVID+ patients are mainly located in the liver, does it mean that it is necessary to conduct in-depth research specifically on the liver to explore the drivers of systemic pathogenesis caused by severe COVID-19?

The importance of the liver as a major driver of regulated proteins in COVID-19 cannot be overstated, given the strong acute phase response induced by SARS-CoV-2 infection and supported by extremely elevated levels of hepatic proteins like CRP, fibrinogen, complement and coagulation factors. We support the suggestion by this reviewer that more research on hepatic involvement in COVID-19 is warranted, and while outside the scope of the present study, future studies should investigate whether hepatic perturbation in COVID represents a generalized response to systemic inflammation caused by the virus versus something more intrinsic to (and directly caused by) the virus itself. This point has been included in the revised discussion (within the study limitations paragraph) as an area for future study: “In addition, studies investigating whether the predominance of alterations in hepatic analytes reflects a generalized inflammatory response induced by SARS-CoV-2 versus a more direct effect of the virus on the liver are warranted.”

7、 Line 333-470: All of these experiments use the samples in Table S6, but not all of them seem to be selected in each experiment. Can the authors describe the principles of sample selection for each experiment in more detail? And why are the samples in Table S1 no longer used in these experiments? If possible, it is recommended that the same samples be used for all experiments for better comparability.

We agree that using samples from the same patients in all assays would be ideal; however, the multiomic studies relied on residual clinical samples in the early days of the pandemic that were unable to be used in later microfluidics experiments. Specifically, the multiomic studies were meant to be hypothesis-generating and were performed well-ahead of the microfluidics studies, as the data was then used to guide more focused mechanistic studies. Unfortunately, a major limitation of using residual clinical samples is the sample volume. While we were able to perform all multiomic (proteomic, lipidomic, metabolomic) analyses on plasma from the same cohort, we did not have remaining sample for the microfluidic assays. Moreover, the multiomic study samples contained EDTA as the anticoagulant, which is not compatible with the microfluidics assays, as EDTA is toxic to endothelial cells. The microfluidic assays are performed under physiologically-relevant flow conditions over time, thus requiring relatively large amounts of plasma with citrate as the anticoagulant. For these reasons,

prospectively collected samples were obtained for the microfluidic assays. Supplemental table 6 outlines the patient demographics for all of the samples used in the microfluidic assays, although not all microfluidic platforms were able to be assessed for every sample. For example, because citrated tubes use a liquid anticoagulant, if the collection tube was underfilled then the red blood cells were still able to be used for single cell assays, however the plasma component would be artificially diluted and thus not valid for use in the other microfluidics experiments. Further, some of the experiments require same day use of the collected sample to avoid any changes in RBC behavior that may occur from sample storage, while plasma could be stored for use in the ELISA assays or the endothelialized device experiments. Even with the benefit of plasma storage technical issues occasionally interfered with aggregation studies, like when an endothelialized device would fail because of inefficient seeding or bacterial contamination. Thus, while using the exact same patient samples for each of these assays would be ideal, the large volumes and experimental timing did not permit doing so. We nevertheless feel confident in our data and conclusions given sufficient power for determining significance with the various platforms used.

8、 Line 333-357: The results shown in Figure3BCD don't seem to be able to tell the RBC aggregation difference between the plasma from COVID-19 patients and critically ill COVID-19-negative patients. Although Figure3E may give a new finding that COVID-19 patients tend to have a wider range of DCVC velocities on an individual, the differences are not significant. Does it mean that although there are different pathways for COVID-19 sepsis, the main pathogenesis is similar to non-COVID-19 sepsis?

Thanks for raising this question so that we can clarify. We agree that the differences in the mean RBC cluster size and velocity are small between our patients with COVID-19 and those with non-COVID-19 sepsis, and that there is likely a lot of overlap in the pathology between the two conditions. Patients with sepsis, regardless of the underlying pathogen, represent a very heterogeneous group with the clinical response and outcome dependent on many factors that are unique to the individual host and infection timing. We believe that in considering the significance of our findings it is important to also consider the timeline of the natural history of both COVID-19 and non-COVID-19 sepsis. For example, management of bacterial-mediated sepsis is driven by antimicrobial therapy which can quickly remove the inciting agent for the inflammatory cascade. By comparison, patients with COVID-19-induced sepsis experience ongoing inflammation as seen in studies examining the persistent high inflammatory markers in patients with severe COVID-19. Therefore, the ongoing exposure to aggregates which vary in both size and velocity could potentiate the persistent endothelial damage in patients with COVID-19, representing a more chronic state (days to weeks) in COVID as compared to more acute (hours to days) in non-COVID sepsis.

9、 Line 426-428: It seems that the authors do not give how the concentrations of these markers are determined.

Thanks for bringing this to our attention so that we can include this information in the revision. Syndecan-1 and vWF were both assessed by ELISA following the manufacturer's instructions. The manuscript text has been modified to include the methodology (ELISA) in the results section, and more detailed information is reported in the dedicated methods section.

10、 Line 457-460: Why are large, freshly collected citrated samples necessary in microfluidics experiments?

The working volumes of our microfluidics experiments include not only the volume of the device itself, but also the transit tubing from the syringe pump and the contents of the syringes themselves. Experiments involving endothelialized devices in particular require even more volume as the transit tubing needs to connect from the syringe pumps to inside the CO2 incubator. The citrate anticoagulant is essential to these experiments as EDTA is toxic to endothelial cells. In addition, in prior experiments involving isolated red blood cells to examine deformability (not included in this publication) and using non-citrate anticoagulant (ie EDTA) we have observed inconsistencies in RBC membrane appearance and behavior in the devices that make the results difficult to interpret. For these reasons we use freshly collected citrated samples to provide the most consistent, reproducible and therefore interpretable data.

11、 Line 506-508: No significant increase in the abundance of fibrinogen chains is detected in acute COVID+ or MIS-C children. Are there other markers that cause RBC aggregation or endothelial glycocalyx degradation found?

This is an important consideration and is also related to point 5 above. We agree with the reviewer that there may be other mediators that promote RBC aggregation outside of the fibrinogen pathway we focused on in our study. As described above in our response to point 5 above, antibodies, particularly IgM, may cause linear RBC aggregation resulting in rouleaux, which we did not directly measure in our cohorts. As our multiomic analyses of the pediatric COVID+ and MIS-C cohorts did not highlight the fluid shear stress pathway, as it did in adult COVID patients, we did not perform additional analyses to look for biomarkers indicative of RBC aggregation or endothelial glycocalyx degradation in children.

12、 Line 547-550: With the development of mass spectrometry, there are many high-throughput methods for the identification of unsaturated fatty acids, such as the combination method of Waters ACQUITY UPC2 and PURSPEC Ω Analyzer. These methods may be helpful to the author's further research.

We appreciate the reviewer for bringing this to our attention. While these methods offer great insight into the composition of lipids, we relied heavily on the expertise and instrumentation present in the Emory Integrated Metabolomics and Lipidomics Core for analysis of analytes. We have shared this new instrumentation with the director of that core and asked that they consider making these platforms accessible.

13、 Line 578-582: In Figure 6, the author focuses on the differences between children and adult patients. Are there some common markers for children and adult patients of severe COVID-19?

We appreciate the reviewer's suggestion to highlight common markers of severe acute COVID in pediatric and adult patients. Our proteomics data demonstrates strong predominance of alterations in complement and coagulation pathways in both adult and pediatric patients; yet, overall, there was surprisingly minimal overlap between our adult and children acute COVID cohorts for altered analytes,

likely related to the fact that the adult cohort highlighted differences between critically ill COVID versus critically ill non-COVID patients, while the pediatric acute COVID cohort was assessed in comparison to healthy children or those with MIS-C. To address this comment we have revised Supplemental Figure 4 to include a Venn diagram of the common overlapping (shared) analytes, with additional details provided as Supplemental Table 19.

Supplemental Figure 4. Pediatric analyte analysis. A.) Table depicting the parameters of our pediatric cohort mass spectrometry run from number of proteins to the percentage of cleavages and average mass error (Δ M) (parts per million (PPM)). B.) & C.) Proteins identified that trend with disease and could be diagnostic for alterations in MIS-C and COVID in children. D.) Venn diagram of the significantly altered proteins (FDR <0.01), lipids & metabolites (P value <0.05) when comparing the adult cohort (COVID+ versus non-COVID) and the pediatric cohort (COVID+ vs Healthy).

14、 Line 659-662: Why can the author come to such a conclusion that the destruction of the protective glycocalyx layer is not a result of enzymatic degradation?

Thanks for asking this question so we can better clarify our conclusions. We did not mean to suggest that circulating sheddases are not involved in the pathophysiology of either COVID-19 or non-COVID sepsis but rather that our experiments are performed on a timeline that is much shorter (30 minutes) than previous studies looking at endothelial glycocalyx degradation as a result of enzymatic activity (2+ hours). We believe that this supports our conclusion that the degradation we observed is related to the mechanical disturbance caused by red cell aggregation, and have modified the text to include the following: “Therefore, to minimize the impact of enzymatic degradation in our model, we designed the assays using a time scale on which biomechanical effects on the glycocalyx could be assessed before any contribution of enzymatic degradation would be expected.”

15、 Line 1240-1380: In this work, a total of three microfluidic chips are used, and the chips used for RBC Deformability and Glycocalyx Degradation Research are very similar in structure, which leads me to mistakenly think that only two chips were used when reading the main text. In order to understand this part better, it is suggested that references should be given when the corresponding microfluidic chips

are first mentioned in the main text (Figure 2C, Figure 2G and Figure 3F). And the structure diagram of each chip should be given in the Supplemental Literature.

Thank you for this suggestion to help clarify the microfluidics experimental setup for the readers. We have added an additional Supplemental Figure (below, new Supplemental Figure 2) and referenced each of the corresponding devices as they are mentioned in the body of the manuscript.

A

B

C

16、 Line 1315: It should be Figure 3E instead of Figure 3D.

Thank you for catching this typo, which we have corrected in the revised manuscript.